

# Rapid simulation of wave runup on morphologically diverse, reef-lined coasts with the BEWARE-2 meta-process model

Robert McCall[1], Curt Storlazzi[2], Floortje Roelvink[1], Stuart G. Pearson[3], Roel de Goede[1], and José A.Á. Antolínez[3]

[1]Deltares, Unit of Marine and Coastal Systems, Boussinesqweg 1, Delft, the Netherlands
[2]U.S. Geological Survey, Pacific Coastal and Marine Science Center, 2885 Mission St, Santa Cruz, California, United States of America
[3]Department of Hydraulic Engineering, Faculty of Civil Engineering and Geosciences, Delft University of Technology, Stevinweg 1, Delft, the Netherlands

**Correspondence:** Robert McCall (robert.mccall@deltares.nl)

**Abstract.** Low-lying, tropical coral reef-lined coastlines are becoming increasingly vulnerable to wave-driven flooding due to population growth, coral reef degradation, and sea-level rise. Early-warning systems (EWS) are needed to enable coastal authorities to issue timely alerts and coordinate preparedness and evacuation measures for their coastal communities. At longer time scales, risk management and adaptation planning require robust assessments of future flooding hazard considering uncer-

tainties. However, due to diversity in reef morphologies and complex reef hydrodynamics compared to sandy shorelines, there have been no robust, analytical solutions for wave runup to allow the development of large-scale coastal wave-driven flooding EWS and risk assessment frameworks for reef-lined coasts. To address the need for a fast, robust prediction of runup along reef-lined coasts, we constructed the BEWARE-2 (Broad-range Estimator of Wave Attack in Reef Environments) meta-process modeling system. We developed this meta-process model using a training dataset of hydrodynamics and wave runup computed

by the XBeach Non-Hydrostatic+ process-based hydrodynamic model for 440 combinations of water level, wave height, and wave period on 195 morphologically diverse representative reef profiles. In validation, the BEWARE-2 modeling system produced runup results that had a root-mean square error of 0.63 m and bias of 0.26 m, relative to runup of 0.17–20.9 m simulated by XBeach Non-Hydrostatic+ for a large range of oceanographic forcing conditions and for a diverse reef morphologies. Incorporating parametric modifications in the modeling system to account for variations in reef roughness and beach slope allows

systematic errors (relative bias) in BEWARE-2 predictions to be reduced by a factor of 1.5–6.5 for relatively coarse or smooth reefs, and mild or steep beach slopes. This prediction is provided by the BEWARE-2 modeling system 4–5 orders of magnitude faster than the full, process-based hydrodynamic model and could therefore be integrated in large-scale EWS for tropical, reef-lined coasts, as well as used for large-scale flood risk assessments.

## 1   Introduction

Low-lying, tropical coral reef-lined coastlines are vulnerable to wave-driven overwash and flooding. These flooding events will become more frequent as sea level rises due to climate-change (Vitousek et al., 2017; Vousdoukas et al., 2018), threatening





infrastructure, ecosystems, and freshwater resources and infrastructure (Reynolds et al., 2015; Shimozono et al., 2015; Albert et al., 2016; Storlazzi et al., 2018). Globally, coral reefs protect more than 200,000 people living on the coast from flooding annually (Beck et al., 2018). These populations are denser, growing faster, and composed of more people from lower-middle

income groups than the global average (Kumar and Taylor, 2015; Sing Wong et al., 2022). The combined increase in hazard probability and exposure is leading to a higher likelihood of flood-related deaths (Chilunga et al., 2017). This increased risk necessitates: 1) better science and understanding of flood risk drivers in the tropics; 2) better, locally-appropriate risk reduction and adaptation strategies to reduce coastal flooding and associated hazards (Hinkel et al., 2014); and 3) access to warning systems to increase preparedness and implement flood mitigation actions (Winter et al., 2020).

Short-term forecasts (typically up to 5–7 days) produced by EWS allow authorities to issue timely warnings and coordinate preparation and evacuation measures, which ultimately reduce risk to lives and assets. Events such as Typhoon Haiyan (Roeber and Bricker, 2015) and widespread flooding of islands due to remotely generated swell in the western Pacific (Hoeke et al., 2013) highlight the need for these warnings of impending coastal flooding events. One third of the world population live in areas that are not covered by EWS, with particularly low EWS coverage in least developed countries and small island

developing states (WMO, 2022a). This issue has gained worldwide attention, and as a result the United Nations-endorsed Sendai Framework for Disaster Risk Reduction has called for improved access to early warning systems and disaster risk assessments by 2030 (UNISDR, 2015).

EWS have been implemented in sandy coastal environments (e.g., Doong et al., 2012; Coz et al., 2021; Stockdon et al., 2023), but reef-lined coasts demand different modeling approaches due to the complex bathymetry and wave dynamics char-

acteristic of reef environments. Coral reef-lined (fringing reef) coasts are typically fronted by a shallow reef flat and a steeper fore reef farther offshore that dissipates open-ocean incoming wave energy through wave breaking and bottom friction (Lowe et al., 2005; Monismith et al., 2013). Wave breaking on the fore reef induces high radiation stress gradients, which results in significant wave-induced set-up on the reef flat (Pomeroy et al., 2012; Beck et al., 2018). Some of the dissipated incident-band (sea-swell; > 0.05 Hz) wave energy is transferred to infragravity-band ("IG," 0.005–0.05 Hz) and very low frequency-band

waves ("VLF," 0.001–0.005 Hz) through breakpoint forcing (Symonds et al., 1982) on the steep fore reef (Péquignet et al., 2009; Pomeroy et al., 2012; Péquignet et al., 2014; Cheriton et al., 2016). Resonant amplification of IG and/or VLF waves can occur when their energy is concentrated at the natural frequency of the reef flat, which is most likely to occur on smooth reefs, with increasing water depth, high incident-band wave periods, and specific reef dimensions (Péquignet et al., 2009, 2014; Quataert et al., 2015; Gawehn et al., 2016; Pearson et al., 2017; Buckley et al., 2018). The complex interaction between tides,

storm surge, wave-induced set-up, incident-band waves, IG waves, and VLF waves drives runup and subsequent flooding along reef-lined coasts.

Findings from sandy beach investigations are not necessarily transferable to coral reef-lined coasts. First, the bathymetric profiles of reef-lined coasts are starkly different from those of gently sloping beaches (Scott et al., 2020). Secondly, coral reefs can have much greater bed roughness, varying between the extremes of relatively smooth "pavement" reefs with little

coral coverage to rough regimes with high coral coverage or/and bedrock rugosity (Quataert et al., 2015; Harris et al., 2018). Together, these severely limit the use of parametric models (e.g., Stockdon et al., 2006; Merrifield et al., 2014) in coral





reef environments (see also Astorga-Moar and Baldock, 2023). Processes-based models (e.g., Roelvink et al., 2009, 2018) have been adapted for coral reef-lined coasts by modifying the typical wave action and/or non-linear shallow water equations and parameterizing the hydrodynamic roughness in wave and friction factors (Van Dongeren et al., 2013; Quataert et al.,

2015; Buckley et al., 2018; Lashley et al., 2018; de Ridder et al., 2021). However, although accurate, these models are very computationally expensive (e.g., Quataert et al., 2020) and therefore too slow for EWS (Winter et al., 2020; WMO, 2022b). To capture the accuracy of process-based models in operationally feasible computational time frames, surrogate models, including metamodels (Pearson et al., 2017; Rueda et al., 2019; Liu et al., 2023) and machine-learning models (Franklin and Torres-Freyermuth, 2022), have been developed by running process-based models over a limited number of schematic coral reef

bathymetries. However, as demonstrated by Scott et al. (2020), the natural variability in coral reef widths, depths, slopes, and rugosities (bathymetric variability) far exceeds the limited schematic bathymetries used in current surrogate models, limiting their accuracy and global applicability in EWS.

To address this need for a fast, accurate EWS for tropical, reef-lined coasts, we developed the Broad-range Estimator of Wave Attack in Reef Environments (BEWARE-2), a computationally efficient meta-process modeling system that estimates

runup (wave-driven set-up and swash) based on complex, process-based hydrodynamic model simulations. Here we first detail the creation of a database of representative reef profiles and the matching of real-world reef profiles to those representative profiles. Next, we document the application of the process-based hydrodynamic model XBeach Non-Hydrostatic+ (de Ridder et al., 2021) to the representative reef profiles over a broad range of oceanographic forcing conditions. Then we describe the application of the meta-model to compute runup. Subsequently, we address meta-model validation and skill quantification.

Lastly, we discuss the meta-process model benefits, limitations, application, and next steps.

## 2 Methods

The BEWARE-2 meta-process modeling system is based on the non-hydrostatic version (de Ridder et al., 2021) of the open-source, process-based hydrodynamic model XBeach (Roelvink et al., 2009, 2018), henceforth referred to as XBNH, to estimate wave runup (wave-driven setup and swash) on reef-lined coasts. To do so, we developed a training dataset of hydrodynamics

and wave runup computed by XBNH for varying hydrodynamic forcing conditions, similar to that of Pearson et al. (2017), for a set of morphologically diverse reef profiles, based on work by Scott et al. (2020), which is described in the following sections. Here we illustrate the different steps in the development of the meta-process modeling system.

### 2.1 Model training dataset

#### 2.1.1 Representative reef profiles

A database of 195 representative, shore-normal, cross-reef profiles was created by combining 175 representative reef profiles from Pearson et al. (2017) and Scott et al. (2020), as well as a set of 20 wide reef profiles (defined as profiles that reach a depth of 15 m at distances greater than 1.5 km offshore) identified in this study (Figure 1). Coral reef profiles included in





the BEWARE-1 model (Pearson et al., 2017) comprise 30 parametric representations of coral reef profile geometries found in literature from 10 sites around the world (Quataert et al., 2015), with reef flat widths ranging from 0–500 m and fore reef slopes from 0.1–0.5. The reef profiles of Scott et al. (2020) and the wide reef profiles, were extracted from a dataset of 30,166 coral reef profiles (Storlazzi et al., 2019), covering the coral reef-lined coasts of the United States of America (U.S.), including the States of Hawai'i and Florida; the Territories of Guam, American Samoa, and U.S. Virgin Islands; and the Commonwealths of Puerto Rico and the Northern Mariana Islands. The dataset consists of transects with a 2 m cross-shore resolution spaced at 100 m intervals along the 3,300+ kilometers of coastline. Scott et al. (2020) removed 9,712 wide reef profiles from the dataset due to inherent uncertainties in the application of XBNH on these profiles, and reduced the remaining 20,454 reef profiles to 500 cluster groups and representative profiles using data reduction techniques on morphology and hydrodynamics of the reef profiles. The 20 wide reef profiles selected in this study were derived from the 9,712 wide reef profiles excluded by Scott et al. (2020), following application of the same using morphological clustering technique.

For the development of BEWARE-2, a set of 175 representative reef profiles (RRPs) was developed based on the 30 initial representative reef profiles (iRRPs) of Pearson et al. (2017) and 500 iRRPs of Scott et al. (2020), neither of which contain any very wide coral reef profiles, and are henceforce termed the 530 normal iRRPs. In general, these 530 iRRPs have shapes characteristic of atoll and fringing reef profiles. Following the methodology of Scott et al. (2020), the 530 normal iRRPs were reduced to 175 normal RRPs by means of hierarchical clustering based on two sets of features, namely the submerged morphology and nearshore hydrodynamics, that were assigned equal weighting. In line with Scott et al. (2020), the submerged morphology was expressed through both the profile depth and the inverse wave celerity, where the wave celerity (calculated from linear wave theory assuming a peak wave period $T_p$ of 8 s) was included to give greater weight to shallower areas. The nearshore hydrodynamics features included the wave-driven setup, sea-swell and IG swash, and 2%-exceedance runup ($R_{2\%}$), computed using XBNH for four representative wave conditions. Similarly following Scott et al. (2020), each feature within each subset was transformed using a Min-Max linear scaler and equal weighting within the subset was assigned during clustering. During hierarchical clustering, iRRPs were progressively merged based on the smallest intergroup dissimilarity. After merging, an RRP was assigned for each group based on its proximity to the mean $R_{2\%}$. If there were more than one RRP with equal distance to the mean, the RRP that represented the larger number of profiles from the initial round of clustering was chosen. The number of cluster groups was set at 175 profiles based on the intra-cluster similarity of relevant hydrodynamic parameters, which resulted in a relative difference of less than 10% for all four parameters.

To allow for the application of BEWARE-2 in areas with wide reefs, such as barrier or extremely wide fringing reefs, additional clustering was performed on the 9,712 reef profiles excluded by Scott et al. (2020) from the dataset of Storlazzi et al. (2019). These profiles were clustered based on the submerged morphology (profile depth and inverse wave celerity) in an identical fashion to that of Scott et al. (2020). Twenty geometrically distinct, wide reef iRRPs were identified and directly mapped to 20 wide reef RRPs. These were added to the 175 normal RRPs for a total dataset of 195 RRPs (Figure 2).







**Figure 1.** Flow chart outlining datasets and processes to develop both the 550 intermediate (iRRP) and final 195 representative reef profile (RRP) databases encompassing the 30 schematic reef profiles of Pearson et al. (2017), the 20,454 narrow reef profiles of Scott et al. (2020), and the 9,712 wide reef profiles excluded from the Scott et al. (2020) analysis but included here.

**2.1.2 XBNH process-based model description and set-up**

XBNH is a phase-resolving, non-hydrostatic model for the nearshore and coast. The model solves the non-linear shallow water equations in a reduced 2 vertical-layer system that includes a non-hydrostatic pressure term, allowing application in shallow to intermediate water depth ($kh \lesssim 4$, where $k = \frac{2\pi}{L}$ is the wave number, $h$ is the water depth, and $L$ is the wave length) with minimal dispersion errors (de Ridder et al., 2021). XBNH and the earlier one-layer non-hydrostatic version of XBeach (Smit 125 et al., 2010) have previously been shown to reproduce laboratory and field measurements of wave transformation and runup on







**Figure 2.** Overview of the morphology of the 195 representative reef profiles (RRPs). The RRPs are color-coded according to projected runup, with yellow indicating profiles with low resulting runup and black those with high resulting runup. In general, the RRPs with narrower and steeper shallow ($< 5$ m depth) portions of the profile have greater resulting runup.

reef profiles well (e.g., Pearson et al., 2017; Lashley et al., 2018; Klaver et al., 2019; Masselink et al., 2019; Pomeroy and van Rooijen, 2019; Quataert et al., 2020; Masselink et al., 2020; de Ridder et al., 2021; Laigre et al., 2023).

XBNH models used for the training and validation of BEWARE-2 were set up in one-dimensional (i.e., cross-shore transect) mode with spatially varying grid resolution and bed roughness. The grid resolution was optimized according to the local water depth and wave period, with a coarser grid being used at greater depths and for longer wave periods (64 points per local wave length; minimum and maximum grid resolution of 0.25 and 5 m, respectively). The reef roughness was parametrized following Storlazzi et al. (2019), with a friction value ($c_f$) of 0.003 for the sandy beach and deep water. An increased reef friction




($c_f = 0.05$) is implemented at all depths at which $kh \leq 1$ (i.e., depth less than 6.8–91.6 m, depending on the wave period

given in Table 1) up to a minimum depth of 0.5 m. The offshore boundary location was established for each individual forcing

condition, taking into account depth restrictions of $n = \frac{c_g}{c} < 0.75$, where $c_g$ is the group velocity and $c$ is the wave celerity, to

ensure correct infragravity wave boundary condtions, and $kh < 4$ to allow for correct dispersion of the incident-band waves.

Additionally, the maximum wave height to depth ratio ($\frac{H_{s,0}}{h}$, where $H_{s,0}$ is the deep water significant wave height) was set to

0.25 to prevent wave breaking at the boundary. Resulting offshore water depths vary between 30 and 58 m. Where necessary,

the profiles were extended from the original RRP depth of 30 m to the offshore water depth required for the forcing conditions

using a 1/10 artificial slope. To enable computations for the full range of hydrodynamic conditions without runup exceeding

the profiles' beach crest, an artificial semi-infinite beach was created extending from 0–30 m MSL with a slope of 1/10. Key

model parameters were adopted from Quataert et al. (2020), with model wave breaker parameters *maxbrsteep* and *reformsteep*,

which control the onset and cessation of wave breaking in the non-hydrostatic model, set to of 0.6 and 0.3, respectively, and

other model parameters set to their default value.

### 2.1.3   XBNH simulations and output

The XBNH models were developed to simulate wave runup on each RRP for 440 combinations of offshore still water level

(SWL; 0–4 m + MSL), $H_{s,0}$ (1–11 m), and $T_p$ (6–22 s), see Table 1 for values applied in this study. To avoid the use of

unrealistic wave conditions, $H_{s,0}$–$T_p$ combinations with a deep water wave steepness ($s_0 = \frac{2\pi H_{s,0}}{gT_p^2}$, where $g$ is the gravitational

constant) greater than 0.075 were removed. Consequently, no $H_{s,0}$ greater than 4 and 7 m were applied in combination with $T_p$

of 6 and 8 s, respectively. Time series of random realizations of incident-band and wave group-bound IG waves were generated

internally by the model (see  Roelvink et al., 2009) for each SWL–$H_{s,0}$–$T_p$ forcing combination assuming normally incident

offshore waves and a JONSWAP spectral shape (peak enhancement factor $\gamma = 3.3$; Hasselmann et al. 1973) and approximately

24° directional spread ($s = 10$ in $cos^{2s}$ directional spread model). Simulations on all RRPs used the same time series of

random incident waves per SWL–$H_{s,0}$–$T_p$ combination to allow for direct comparison of hydrodynamics between RRPs. All

simulations were run for a period of 800 $T_p$ (i.e., approximately 800 waves), of which the first 300 $T_p$ period was used as model

spin up and model output from the final 500 $T_p$ was used for runup analysis.

| Forcing parameter | Value |
|---|---|
| SWL (m + MSL) | 0, 1, 2, 3, 4 |
| $H_{s,0}$ (m) | 1, 2, 3, 4, 5, 6, 7, 8, 9, 10, 11 |
| $T_p$ (s) | 6*, 8*, 10, 12, 14, 16, 18, 20, 22 |

**Table 1.** Overview of hydrodynamic forcing conditions used in the training dataset. *Note: wave height combinations for $T_p \leq 8$ s are
constrained by maximum deep water wave steepness ($s_0 \leq 0.075$).

The XBNH models were set to output the time series of the horizontal and vertical position of the simulated waterline at a

temporal resolution of $\frac{T_p}{20}$ (i.e., 0.3–1.1 s). The value of $R_{2\%}$ was derived from the vertical waterline position time series follow-





ing Stockdon et al. (2006). Output time series for each simulation were split into five time frames of 100 $T_p$ (i.e., approximately
100 waves) to provide five estimates of the empirical $R_{2\%}$ values per simulation. In a limited number of simulations, waterline
position time series were split into four or three (6.2 and 2.3% of simulations, respectively) time frames to ensure there were
sufficient runup maxima in each period to determine the empirical $R_{2\%}$ value.

In addition to the time series of the simulated waterline, XBNH output time series of water levels, depth-averaged velocity
and horizontal discharge were stored at the offshore boundary of the model, eight locations across the reef profile, and 15
positions on the beach face at a temporal resolution of $\frac{T_p}{20}$, which may in the future be used to estimate nearshore wave heights
and overtopping rates. These data, although provided in the BEWARE-2 database (https://doi.org/10.5066/XXXXXXXX), are
not discussed further in this paper.

## 2.2    Meta-process model description

Application of BEWARE-2 to estimate wave runup for a given real-world "target" reef profile and real-world "target" oceanic
boundary conditions follows a three-step process (Figure 3). These three steps are discussed in the following sections.

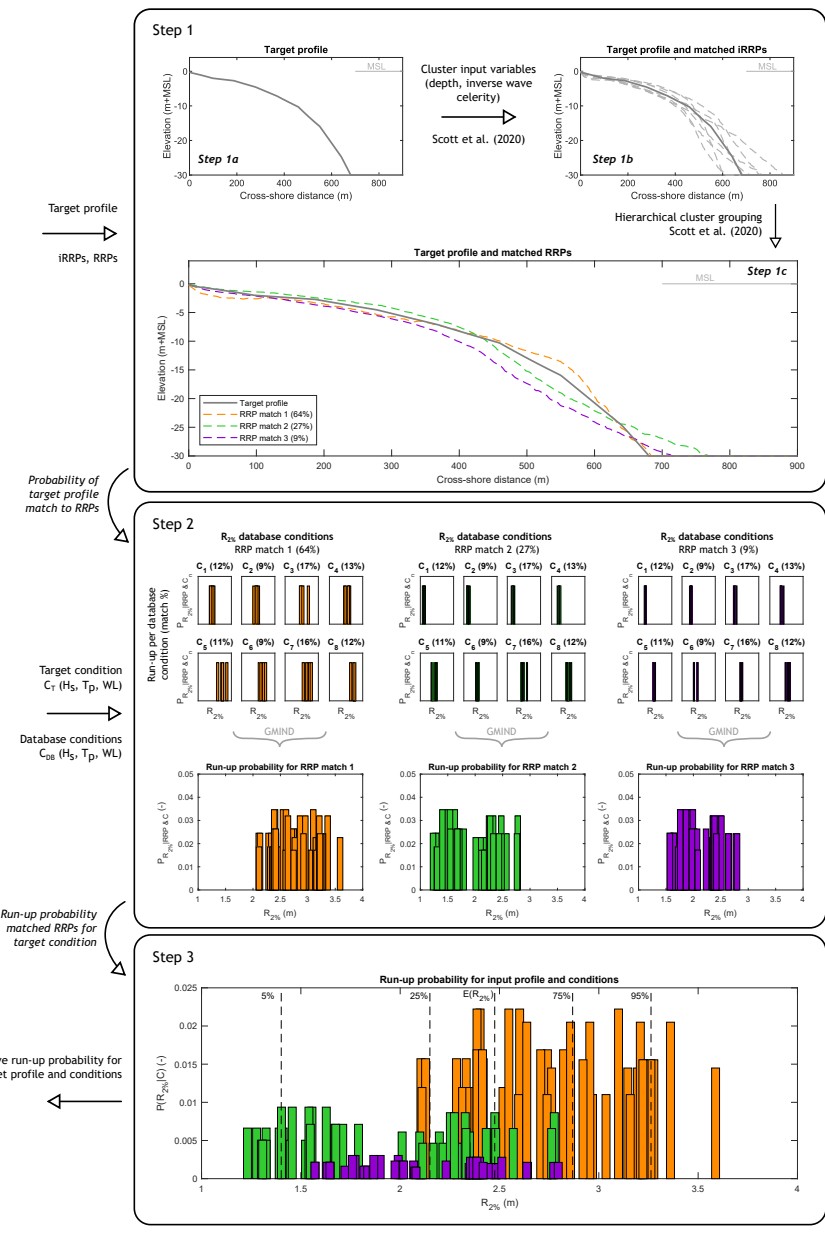

**Figure 3.** Schematic displaying the steps in application of BEWARE-2 to compute a runup, in this case using three matched representative reef profiles (RRPs) to the real-world, "target," bathymetric profile. Step 1: Matching the target profile to RRPs and their relative match probability. Step 2: For each RRP, extract runup values associated with the oceanographic forcing conditions in the BEWARE-2 database bounding the target oceanographic forcing conditions and their probabilities relative to the target forcing oceanographic conditions using the geometric mean inverse normalised distance (GMIND), Step 3: Combine the database runup probabilities associated with each RRP and their relative weighting to the RRP to compute an expected and/or exceedance probability runup.





### 2.2.1 Step 1: Matching target bathymetric profile to representative reef profiles (RRPs)

Probabilistic matching of a real-world, "target" bathymetric profile to the representative reef profiles (RRPs) involves three steps (Figure 3, top panel). First, the cluster input variables (i.e., depth and inverse wave celerity) are generated (Step 1a). Second, the target profile is probabilistically matched to the 550 iRRPs (Step 1b). Finally, the result of the matching to the 550 iRRPs is transformed to the 195 RRPs through the hierarchical tree (Step 1c).

The cluster input variables for the probabilistic profile matching in Step 1a are computed following the same procedure for the submerged morphology clustering as Scott et al. (2020), see Section 2.1.1. For both the target profile and the 550 iRRPs, the clustering variables are weighted by 50% for the profile depths and 50% for the inverse wave celerities.

The distance between the target profile and the 550 iRRPs in Step 1b is calculated using the City Block Distance metric (e.g., Melter, 1987). The softmax function (Bridle, 1990) subsequently transforms the vector of real values, in this case the normalized cluster parameter distances, and a stiffness parameter ($\beta_{\text{softmax}}$) into matching probabilities of the target profile to the iRRPs ($p_{\text{P,iRRP}}$):

$$p_{\text{P,iRRP},i} = \frac{e^{-\beta_{\text{softmax}} d_i}}{\sum_{j=1}^{N_{\text{iRRP}}} e^{-\beta_{\text{softmax}} d_j}} \tag{1}$$

where $p_{\text{P,iRRP},i}$ is the matching probability of the target profile to iRRP $i$, $d_i$ is the distance between the target profile and iRRP $i$, $d_j$ is the distance between the target profile and iRRP $j$, and $N_{\text{iRRP}} = 550$ is the total number of iRRPs.

The stiffness parameter $\beta_{\text{softmax}}$ can be seen as a concentration parameter controlling entropy. Larger values of $\beta_{\text{softmax}}$ result in narrower distributions, corresponding to higher probabilities associated with small absolute distance metrics between the target profile and the iRRPs (and similarly, low probabilities for large absolute distance metrics). The value of $\beta_{\text{softmax}}$ is initially set to 1200, following Scott et al. (2020). A varying $\beta_{\text{softmax}}$ can improve matching outcomes if entropy is too small or large, i.e., many or few iRRPs profiles match the target profile (Bakker et al., 2022). Thus a varying stiffness criteria is established on the distance metric entropy, and ultimately converted on profile matching threshold numbers. Specifically, if the number of iRRPs that a target profile matches to is less than four, $\beta_{\text{softmax}}$ is stepwise relaxed until four iRRPs are matched. Conversely, if the target profile matches more than 10 iRRPs, $\beta_{\text{softmax}}$ is incrementally tightened, with the additional benefit of improving computational efficiency during interpolation.

Finally, in Step 1c, the matched iRRPs are linked to RRPs following the hierarchical clustering tree (i.e., linking 530 iRRPs to 175 RRPs) in the case of regular reef profiles, and in the case of wide reef profiles, the tree branch is a direct iRRP–RRP matching (see Figure 1). Note that due to hierarchical clustering of iRRPs to RRPs, the minimum number of matched RRPs may be less than four. Where multiple matched iRRPs link to the same RRP through hierarchical clustering, the final probability of profile match between the target profile and the RRP ($p_{\text{P,RRP}}$) is computed by summing the match probabilities of all linked iRRPs.




### 2.2.2 Step 2: Computing runup estimates for matched RRPs based on target oceanic conditions

Wave runup estimates for the "target" oceanic forcing condition $C_T(H_{s,0}, T_p, \text{SWL})$ are generated for each matched RRP (Figure 3, center panel). For each matched RRP, all BEWARE-2 $R_{2\%}$ values associated with the oceanic conditions bounding (higher and lower) the target oceanic condition are identified in the BEWARE-2 database, resulting in $2^3 = 8$ bounding oceanic conditions $C_{DB}$ (all combinations of the nearest greater and lesser database values of $H_{s,0}$, $T_p$ and SWL relative to the target oceanic condition). For example, if the target oceanographic conditions are $H_{s,0} = 4.2$ m, $T_p = 13.7$ s, and SWL = 1.4 m, $R_{2\%}$ values for all combinations of $H_{s,0} =[4, 5]$ m, $T_p =[12, 14]$ s, and SWL =[1, 2] m conditions would be identified in the BEWARE-2 database. Each BEWARE-2 $R_{2\%}$ value is then assigned a weighting factor proportional to the geometric mean inverse normalised distance (GMIND) between the target oceanographic condition and those in the BEWARE-2 database:

$$
\text{GMIND}(n) = \left( \prod_{j=1}^{J} 1 - \left| \frac{X_{j,n} - Y_j}{\Delta X_j} \right| \right)^{\frac{1}{J}}
\tag{2}
$$

where $n = 1, \ldots, 8$ is the index of the eight ($2^J$) nearest oceanic conditions in the database, $J = 3$ is the number of oceanic forcing variables (i.e., $H_{s,0}$, $T_p$ and SWL), $X_{j,n}$ is the value of variable $j$ in the database oceanic condition $C_{DB}(n)$, $Y_j$ is the target oceanic condition for variable $j$, and $\Delta X_j$ is the spacing of variable $j$ in the BEWARE-2 oceanic condition dataset.

The probability weighting ($p_{C_{DB}|C_T}(n)$) between the target condition $C_T$ and each of the eight nearest oceanic conditions in the database ($C_{DB}(n)$) is found by normalising GMIND($n$) for each $C_{DB}$ condition by the sum of GMIND for all eight $C_{DB}$:

$$
p_{p_{C_{DB}|C_T}}(n) = \frac{\text{GMIND}(n)}{\sum_{m=1}^{8} \text{GMIND}(m)}
\tag{3}
$$

### 2.2.3 Step 3: Compute runup estimate for target profile and target oceanic conditions

To compute the overall expected and/or exceedance probability $R_{2\%}$ for a target profile and target oceanic forcing conditions, the $R_{2\%}$ probabilities of Step 2 ($p_{C_{DB}|C_T}(n)$)) are multiplied by the probabilities of profile match of Step 1 ($p_{P,RRP}$; Figure 3, bottom panel). Based on the weighted $R_{2\%}$ values, the expected value and exceedance probability values for $R_{2\%}$ can be established through an empirical cumulative density function.

### 2.3 Accounting for variations reef roughness and beach slope

The BEWARE-2 training dataset is composed of the results of XBNH simulations on 195 RRPs and under varying conditions of SWL, $H_{s,0}$ and $T_p$; geometric and hydrodynamic parameters that were found by Pearson et al. (2017) to contribute most to variations in wave runup. However, the training dataset only contains simulations that have constant values for the hydrodynamic roughness of the reef ($c_{f,\text{ref}} = 0.05$) and slope of the subaerial beach ($\beta_{b,\text{ref}} = 0.10$), which were found by Pearson et al. (2017) to contribute least to variations in wave runup. Despite their lesser significance for wave runup on coral reef-lined





coasts, inclusion of coral reef roughness and beach slope in BEWARE-2 may be relevant for first-order assessments of the

effect of reef health (expressed through $c_f$) and (seasonal) morphological change of the beach (expressed through $\beta_b$) on wave runup. Although it would technically be feasible to include variations in $c_f$ and $\beta_b$ in the training dataset, this would require an undesirable increase in the number of XBNH simulations and computation time (e.g., a nine-fold increase in computational expense for a minimum of three permutations per parameter, or a 25-fold increase for five permutations per parameter). Therefore, an alternative approach is taken to estimate RRP-specific contributions of reef roughness and beach slope to wave

runup.

The methodologies derived to estimate the effect of reef roughness and beach slope variations on wave runup are described in full in Appendix A. In summary, the methodologies utilize a limited dataset of XBNH simulations on 31 reef profiles with varying hydrodynamic parameters and varying values of $c_f$ and $\beta_b$ (Appendix A1) to calibrate physics-based relations for relative increase or decrease of wave runup using known morphological and hydrodynamic parameters. The resulting

relations used to estimate relative changes in wave runup due to variations in reef roughness and beach slope are described in Section 2.3.1 and 2.3.2, respectively. A full description of the derivation and calibration of these relations is given in Appendix A2 and Appendix A3, respectively.

### 2.3.1 Reef roughness

Increased or decreased wave runup $(R_{2\%}^{m,r}$; Equation 4) relative to the standard BEWARE-2 runup estimate $(R_{2\%})$ due to

decreased or increased reef roughness, respectively, is approximated by estimating the relative change in wave height reaching the shore. To do so, a simple wave energy balance model using linear wave theory, the deep water wave conditions, and reef geometry, but excluding bed friction effects, described in Appendix A2, is used to estimate the cross-shore varying near-bed wave orbital $(u_{\mathrm{orb}})$ and wave group $(c_g)$ velocity. This velocity is integrated over the cross-shore extent of the model domain with a hydrodynamic roughness value of $c_f = 0.05$ $(L_{c_f})$, i.e., the cross-shore extent of the coral reef, to provide a first-order

estimate of potential energy loss on the reef due to bed friction $(\Gamma_{\mathrm{reef}}$; Equation 7). The potential energy loss estimate is subsequently scaled by the difference in roughness of the reef $(c_f)$ relative to the reference reef roughness included in the training dataset $(c_{f,\mathrm{ref}} = 0.05$; Equation 6), $H_{s,0}$, the gravitational constant $(g)$, and a calibration coefficient $(\alpha_r)$ to compute a wave runup modification factor $(F_r$; Equation 5).

$$R_{2\%}^{m,r} = R_{2\%} F_r \tag{4}$$


$$F_r = \max\left(1 + \alpha_r \frac{\gamma_r}{\sqrt{g}H_{s,0}} \Gamma_{\mathrm{reef}}, 0\right) \tag{5}$$

$$\gamma_r = \sqrt{\left|\frac{c_f}{c_{f,\mathrm{ref}}} - 1\right|} \tag{6}$$





$$\Gamma_{\text{reef}} = \sqrt{\int_{L_{c_f}} \frac{u_{\text{orb}}^3(x)}{c_g(x)} dx} \tag{7}$$

A full description of the derivation of Equations 4–7, and the calibration of $\alpha_r$ is given in Appendix A2. In summary, separate values of $\alpha_r$ were calibrated for reef profiles with relatively low roughness and relatively high roughness, resulting in $\alpha_{r,\text{smooth}} = 1.16$ for a relatively smooth reef ($c_f = 0.01$), and $\alpha_{r,\text{rough}} = -0.65$ for a relatively rough reef ($c_f = 0.10$). Although the application of these calibrated values is intended to illustrate likely upper and lower limits of wave runup for natural reefs

in poor or good health, interpolation between the calibrated values for specific intermediate values of $c_f$ likely exceeds the accuracy of the methodology and is therefore not recommended.

### 2.3.2 Beach slope

In an approach similar to that for variations in reef roughness, wave runup on steeper or milder beach slopes than those included in the BEWARE-2 training dataset ($R_{2\%}^{m,b}$) can be estimated by applying Equation 8. In this, a linear wave runup correction

factor ($F_b$; Equation 9), inspired by the work of Stockdon et al. (2006), is computed using information in the BEWARE-2 training dataset on the hydrodynamic components of wave runup. Here, $\eta_{\text{surf}}$, $\eta_{\text{swash}}$, $S_{\text{IG}}$, and $S_{\text{inc}}$ are the surf-zone setup, setup inside the swash zone, infragravity-band swash and incident-band swash components of wave runup, respectively, and $\eta_{\text{surf}} + \eta_{\text{swash}} = \eta_{\text{wl}}$ represents the combined time-averaged setup at the shoreline. Following Stockdon et al. (2006), variations in beach slope ($\beta_b$) are assumed to affect $S_{\text{inc}}$, but not $S_{\text{IG}}$. Furthermore, it is assumed that the beach slope affects water level

set-up at the shoreline (Stockdon et al., 2006), but that this can be separated into a surf-zone component, which is assumed to depend only on the reef and shoreface morphology, and a beach slope-dependent swash-zone component. The contribution of beach slope variations to $S_{\text{inc}}$ and $\eta_{\text{swash}}$ is set through the calibration coefficient $\alpha_b$.

$$R_{2\%}^{m,b} = R_{2\%} F_b \tag{8}$$

$$F_b = \frac{\overline{\eta_{\text{surf}}} + \alpha_b \overline{\eta_{\text{swash}}} + \sqrt{\frac{S_{\text{IG}}^2 + (\alpha_b S_{\text{inc}})^2}{2}}}{\overline{\eta_{\text{surf}}} + \overline{\eta_{\text{swash}}} + \sqrt{\frac{S_{\text{IG}}^2 + S_{\text{inc}}^2}{2}}} \tag{9}$$

Although it might be expected that $\alpha_b$ would scale linearly with changes in beach slope, comparison to the calibration data (Appendix A3) shows that linear scaling does not improve the predictive skill of the model. Instead, a heuristic non-linear relation is found between $\alpha_b$ and the change in beach slope (Equation 10):

$$\alpha_b = \left(\frac{\beta_b}{\beta_{b,\text{ref}}}\right)^{\kappa_b} \tag{10}$$





where $\beta_{b,\mathrm{ref}} = 0.10$ is the reference beach slope included in the BEWARE-2 training dataset. The value of $\kappa_b = \frac{1}{\mathrm{e}}$ was calibrated by minimizing the RB of the predicted wave runup in the calibration dataset, as described in Appendix A3. As the non-linear nature of Equation 10 is not fully understood, extrapolation for beach slopes outside the range of the calibration simulations (i.e., $\beta_b = 0.05$–$0.20$) is not recommended.

### 2.4 Meta-process model validation

#### 2.4.1 Validation dataset

The ability of the BEWARE-2 meta-process model to predict $R_{2\%}$ on morphologically diverse reefs and under varying hydro-dynamic forcing conditions is quantified using a validation dataset of 24,000 process-based, XBNH model simulations that are separate from the dataset of simulations used to train the meta-process model. To develop the validation dataset, five normal, and one to two wide, real cross-shore profiles were selected from each of the seven geographic regions (Guam, Saipan-Tinian,

American Samoa, Hawai'i, Florida, Puerto Rico, and the US Virgin Islands) included in the dataset of Storlazzi et al. (2019), for a total of 35 normal and 13 wide reef profiles. Profiles representative of the diversity in morphology of normal reefs in the dataset of Storlazzi et al. (2019, i.e., 20,454 profiles in total, see Section 2.1.1) were selected statistically for each geographic region by first determining the cross-shore profile of the 5, 25, 50, 75, and 95% depth exceedance values, i.e., the depth exceeded by a given percentage of the observed profiles at each cross-shore location (Figure 4, dashed lines). Subsequently,

the nearest real profiles to the 5, 25, 50, 75, and 95% depth exceedance profiles (Figure 4, solid lines) were selected for the validation dataset. Wide reef profiles were similarly selected for each geographic region from the dataset of wide coral reef profiles of Storlazzi et al. (2019, i.e., 9,712 profiles in total, see Section 2.1.1). For all geographic regions except American Samoa, the nearest observed profiles to the 25% and 75% depth exceedance of wide profiles were selected for the validation dataset. In American Samoa, the only wide profile included in the database of Storlazzi et al. (2019) was selected. None of the

48 (35 normal and 13 wide reef) validation profiles were identical to the RRPs included in the training dataset.

The XBNH models were set up for all 48 validation profiles using the same model settings and procedure as described in Section 2.1.2. Each validation profile was forced with the same set of 100 combinations of SWL, $H_{s,0}$ and $T_p$, where values of SWL, $H_{s,0}$ and $T_p$ were randomly sampled between the minimum and maximum values included in the training dataset (Table 1) following independent uniform probability distributions. In similar fashion to the training dataset, no combinations of $H_{s,0}$

and $T_p$ with a deep-water wave steepness greater than 0.075 were included in the validation dataset. Wave spectral frequency and directional spread parameters were kept the same as in the training dataset (Section 2.1.3).

Parameter values for the reef roughness and beach slope are initially kept the same as in the training dataset. Thus, the valida-tion dataset contains 4800 simulations (48 profiles, 100 oceanic forcing conditions) with the trained values for reef roughness ($c_{f,\mathrm{ref}} = 0.05$) and beach slope ($\beta_{b,\mathrm{ref}} = 0.10$). However, to validate the $R_{2\%}$ modification relations for reef roughness and

beach slope described in Sections 2.3.1 and 2.3.2, additional XBNH simulations were carried out for each combination of validation reef profile and oceanic forcing condition in which either the reef roughness ($c_f = 0.01$, or $c_f = 0.10$), or the beach slope ($\beta_b = 0.05$, or $\beta_b = 0.20$) was increased or decreased relative to $c_{f,\mathrm{ref}}$, or $\beta_{b,\mathrm{ref}}$, respectively. These reef roughness and



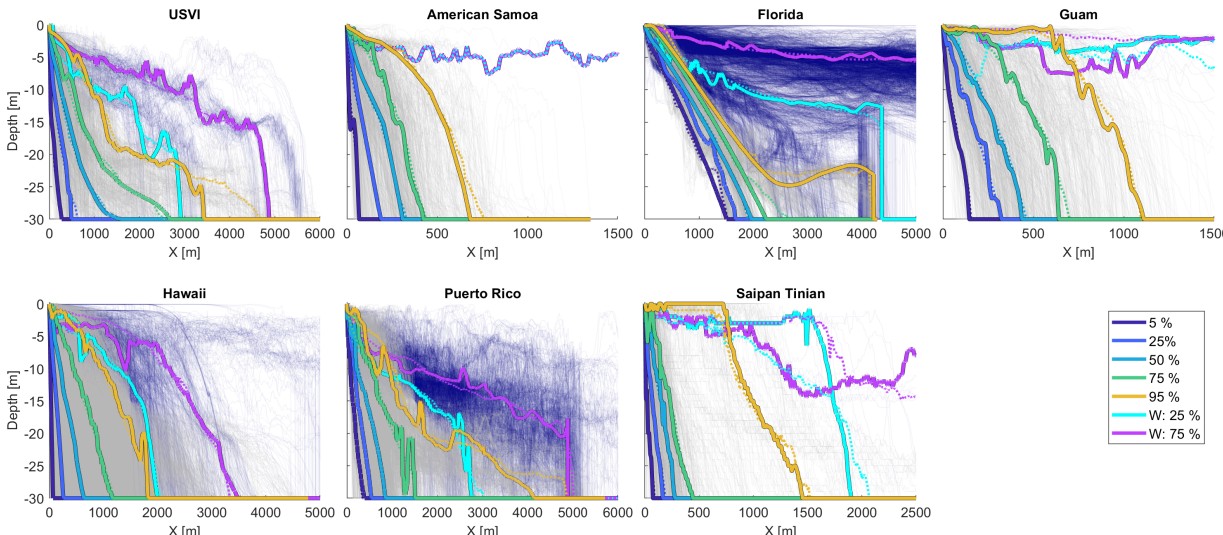

**Figure 4.** Observed, real-world, normal (gray) and wide (dark blue) cross-shore reef profiles for each of the seven geographic regions included in the dataset of Storlazzi et al. (2019). Colored dashed lines indicate the statistical 5, 25, 50, 75, and 95% depth exceedance profiles and colored solid lines indicate their nearest observed profile equivalent used for model validation. For the wide reefs, only the 25 and 75% depth exceedance profiles were extracted for model validation.

beach slope variations compose 19,200 simulations (four variations of roughness and slope for all 4800 reef profile – oceanic forcing condition combination).

The combined 24,000 XBNH validation simulations were run for a period of 800 $T_p$ (of which the first 300 $T_p$ was used as spin up), as was the case for the training dataset. $R_{2\%}$ simulated in the XBNH validation models was derived as described in Section 2.1.3. BEWARE-2 $R_{2\%}$ predictions were calculated using the profile matching and $R_{2\%}$ extraction procedures described in Section 2.2. Where relevant, BEWARE-2 $R_{2\%}$ predictions were modified for the reef roughness and beach slope following the procedures described in Section 2.3.

### 2.4.2 Model skill quantification

The skill of the BEWARE-2 meta-process model in reproducing $R_{2\%}$ simulated by XBNH for the validation dataset was quantified by five measures of accuracy: the root-mean-square error (RMSE; Equation 11), the model bias (Bias, Equation 12), the scatter index, or non-dimensional RMSE (SI; Equation 13), the relative, or non-dimensional bias (RB; Equation 14), and the coefficient of determination (R$^2$, Equation 15).

$$\text{RMSE} = \sqrt{\overline{(X' - X)^2}} \tag{11}$$

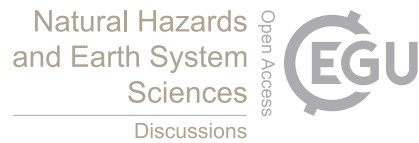

$$\text{Bias} = \overline{(X' - X)} \tag{12}$$

$$\text{SI} = \frac{\text{RMSE}}{\overline{X}} \tag{13}$$


$$\text{RB} = \frac{\text{Bias}}{\overline{X}} \tag{14}$$

$$\text{R}^2 = 1 - \frac{\overline{(X' - X)^2}}{\overline{\left(X - \overline{X}\right)^2}} \tag{15}$$

In Equations 11–15, $X$ represents the $R_{2\%}$ level simulated by XBNH, $X'$ represents the $R_{2\%}$ level predicted by BEWARE-2,
and an overbar represents the arithmetic mean.

## 3 Results

### 3.1 Profile matching

The first step in the BEWARE-2 meta-process model is the probabilistic matching of the target profile to the database RRPs
(see Figure 3). Here, the validation reef profiles (solid colored lines in Figure 4) were probabilistically matched with between
three and 10 RRP profiles ($N_{\text{match}}$; median value 5.5) using the methodology described in Section 2.2 (Figure 5 and Table A1).
In the majority of cases (32 out of 48), the target profile matching included one RRP with at least a 50% probability of matching
($p_{\text{P,RRP}} > 0.5$). In six cases (US Virgin Islands W:25%, American Samoa W, Florida 95%, Guam W:25%, Saipan-Tinian 95%,
and Saipan-Tinian W:25%) the target profile was matched very strongly ($p_{\text{P,RRP}} > 0.9$) to a single RRP. Three cases (American
Samoa 50%, American Samoa 75%, and Puerto Rico 75%) had poor probability of matching ($p_{\text{P,RRP}} < 0.3$) for all matched
RRPs, with the poorest probability of matching occurring for the American Samoa 75% profile (maximum match probability
of 25%).

### 3.2 Runup prediction

Values of $R_{2\%}$ vary substantially between validation profiles (Figure 6), with maximum $R_{2\%}$ on the narrower reef profiles (e.g.,
5% profiles of U.S. Virgin Islands, American Samoa, Hawai'i and Saipan-Tinian) approximately 4–5 times greater than on the
wide reef profiles. BEWARE-2 $R_{2\%}$ predictions (expected value) compare very well with simulated $R_{2\%}$ results of XBNH
almost all 35 normal reef validation profiles, and almost half of the 13 wide reef validation profiles.




**Figure 5.** Probabilistic matching of the 48 validation profiles (thick black lines) to RRPs (colored lines). Probability of matching to each RRP is indicated by the color of the RRP. The number of matched RRPs ($N_{match}$) and maximum matching probability ($p_{max}$) is given below each panel.

For the 35 normal reef validation profiles, RMSE and Bias vary between 0.19–0.93 and -0.18–0.75 m, respectively, whereas SI and RB vary between 0.05–0.21 and -0.04–0.18, respectively (see also Table A1). The coefficient of determination is high ($R^2 \geqq 0.80$) for 31 out of 35 validation profiles, and a minimum of $R^2 = 0.64$ (Hawai'i 95%). The two normal validation profiles that matched very strongly to a single RRP have similar accuracy measures to the overall dataset (Florida 95%: SI = 0.15, RB = 0.12; Saipan-Tinian 95%: SI = 0.08, RB = 0.02). Similarly, the three cases that had poor probability of matching





do not suffer from a substantial reduction in accuracy (American Samoa 50%: SI = 0.08, RB = 0.05; American Samoa 75%: SI = 0.16, RB = 0.14; Puerto Rico 75%: SI = 0.11, RB = 0.08). Overall, there is no significant correlation (Pearson correlation coefficient 0.01; p-value 0.94) between the accuracy of BEWARE-2 (in this case simplified to only SI, but similar results are found for RB) and the highest probability of matching ($p_{P,RRP}$) to an RRP. This suggests that the maximum matching probability to RRPs is not a key factor determining model skill, and that the accuracy of the BEWARE-2 meta-process model will remain broadly similar for all normal reef profiles, independent of their morphological similarity to the RRPs.

For the 13 wide reef validation profiles, RMSE and Bias vary between 0.25–1.17 and -0.81–1.02 m, respectively, whereas SI and RB vary between 0.08–0.43 and -0.16–0.37, respectively. Only 5 of the 13 wide reef validation profiles have a high coefficient of determination ($R^2 \geqq 0.80$). It is therefore apparent that the accuracy of the BEWARE-2 meta-process model is substantially lower on wide reef profiles (e.g., barrier or extremely wide fringing reefs) than on normal reef profiles. Again for the wide reef validation profiles, there is no clear correlation between the accuracy of BEWARE-2 and a high probability matching to RRPs (Pearson correlation coefficient 0.22; p-value 0.47), exemplified by the relatively low accuracy for American Samoa W (one of six cases to match very strongly to one RRP; SI = 0.35, RB = 0.30) and the relatively high accuracy of Hawai'i W:25% (matched to seven RRPs with maximum $p_{P,RRP} = 0.56$; SI = 0.10, RB = -0.05).



**Figure 6.** Values of $R_{2\%}$ simulated by XBNH (horizontal axis) and the expected value prediction of BEWARE-2 (vertical axis) for all oceanic forcing conditions and each validation profile. Figures include the 1:1 relation (solid black line), 10% upper and lower deviation from 1:1 (dashed black lines) and the linear regression through the data (solid orange line).



Model skill metrics for the entire validation dataset (Figure 7, left panel) are generally similar to those of the individual validation profiles, with an RMSE and Bias of 0.63 and 0.26 m, respectively, and SI and RB of 0.13 and 0.05, respectively. Model accuracy is generally higher for the 35 normal reef validation profiles ($R^2$ = 0.98, RMSE = 0.56 m, Bias = 0.26 m, SI = 0.10, RB = 0.05; not shown separately) than for the 13 wide reef profiles ($R^2$ = 0.72, RMSE = 0.77 m, Bias = 0.25 m, SI = 380 0.25, RB = 0.08; see also Figure 8, bottom-right panel).

In addition to the expected value of $R_{2\%}$, BEWARE-2 provides the probability of multiple $R_{2\%}$ estimates, based on the probability of profile matching (Section 2.2.1) and the probabilistic interpolation of oceanic forcing conditions included in the training dataset (Section 2.2.2), which can be used to determine empirical $R_{2\%}$ prediction confidence intervals. The 50% confidence interval prediction of $R_{2\%}$ (i.e., 25–75% $R_{2\%}$ prediction interval) is 26% ±12% (mean and standard deviation) 385 of the expected value of the prediction (Figure 7, left panel). Whereas 16% of XBNH $R_{2\%}$ values in the validation dataset fall exactly within the narrowest 10% confidence interval (i.e., 45–55% $R_{2\%}$ prediction interval), 67% of all XBNH $R_{2\%}$ values fall within the 50% confidence interval prediction, and 92% fall within the 90% confidence interval prediction (Figure 7, right panel). Again, these values are higher when considering only the 35 normal reef validation profiles (18, 75, and 98%, respectively) than when considering the 13 wide reef validation profiles (9, 49, and 75%, respectively). Note that since the 390 BEWARE-2 confidence intervals were determined empirically from information in the training dataset (as well as the profile matching procedure) and applied to the validation dataset, rather than derived directly from the validation data, these results are pertinent and provide valuable information for the application of the BEWARE-2 model.

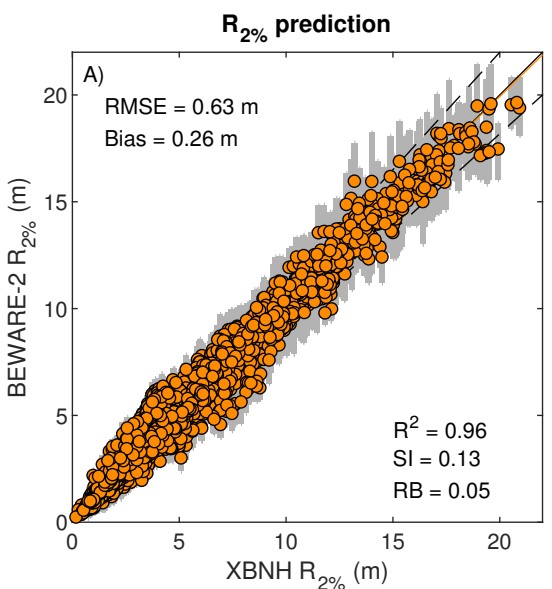

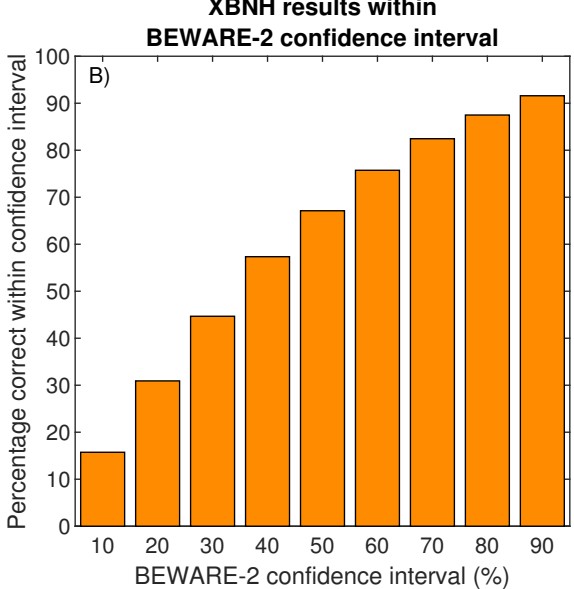

**Figure 7.** Assessment of $R_{2\%}$ prediction and confidence intervals. A) Comparison of $R_{2\%}$ simulated by XBNH (horizontal axis) and the expected value prediction of BEWARE-2 (vertical axis) for all validation simulations. Included are the BEWARE-2 50% prediction confidence interval (gray vertical bands), the 1:1 relation (solid black line), 10% upper and lower deviation from 1:1 (dashed black lines) and the linear regression through the data (solid orange line). Model skill metrics refer to those of the expected value prediction. B) Percentage of XBNH $R_{2\%}$ values matching BEWARE-2 predictions within BEWARE-2 10–90% confidence intervals.

The model skill for the validation cases differs little for variations in oceanic forcing (Figure 8). The relative measures of model skill $R^2$, SI, and RB generally improve marginally for increasing $H_s$ (Figure 8, top row) and for decreasing $s_0$

395 (Figure 8, second row), whereas RB increases marginally for greater SWL (Figure 8, third row). For all variations in oceanic forcing conditions, the difference in accuracy is insubstantial smaller than the overall accuracy measures (i.e., difference in SI is approximately 10–20% of the overall SI) and will thus likely not affect the practical application of BEWARE-2. Variation in profile steepness, however, expressed through $W_{\text{reef}}$ (defined as the cross-shore extent between the shoreline and the 15 m depth contour), again highlights the substantially lower $R^2$ and higher SI values (i.e., lesser accuracy) for very wide reef profiles, compared to those of narrow to moderately wide (i.e., $W_{\text{reef}} < 1500$ m) reef profiles (Figure 8 bottom row).

400

The skill metrics of BEWARE-2 are relatively insensitive to the selection of the minimum number of iRRPs the target profile is matched to. In sensitivity simulations using between two and six matched iRRP profiles at minimum, the median skill metrics for all 49 validation profiles changed by less than 8, 11, and 1%, for SI, RB and $R^2$, respectively. For instance, the median SI value for the 49 validation profiles in the sensitivity simulations ($SI = 0.10$–$0.12$) is less than 8% change relative to the

405 median SI value for the standard case of minimum four matched iRRPs ($SI = 0.11$).



**Figure 8.** Wave runup simulated by XBNH (horizontal axis) and the expected value prediction of BEWARE-2 (vertical axis) for variations in deep water wave height (top row), deep water wave steepness (second row), still water level (third row) and reef width (bottom row). Figures include the 1:1 relation (solid black line), 10% upper and lower deviation from 1:1 (dashed black lines) and the linear regression through the data (solid orange line).





### 3.3 Prediction of reef roughness and beach slope effects

The wave runup computed by XBNH for simulations with increased or decreased reef roughness and beach slope are compared to BEWARE-2 predictions of wave runup without and with the use of Equations 4 and 8 ("Uncorrected" and "Corrected" columns in Figure 9, respectively). Application of these correction factors substantially increases predictive skill of BEWARE-2 for the validation profiles, with the RB decreasing by a factor of 1.5–6.5 compared to BEWARE-2 predictions without correction factors.

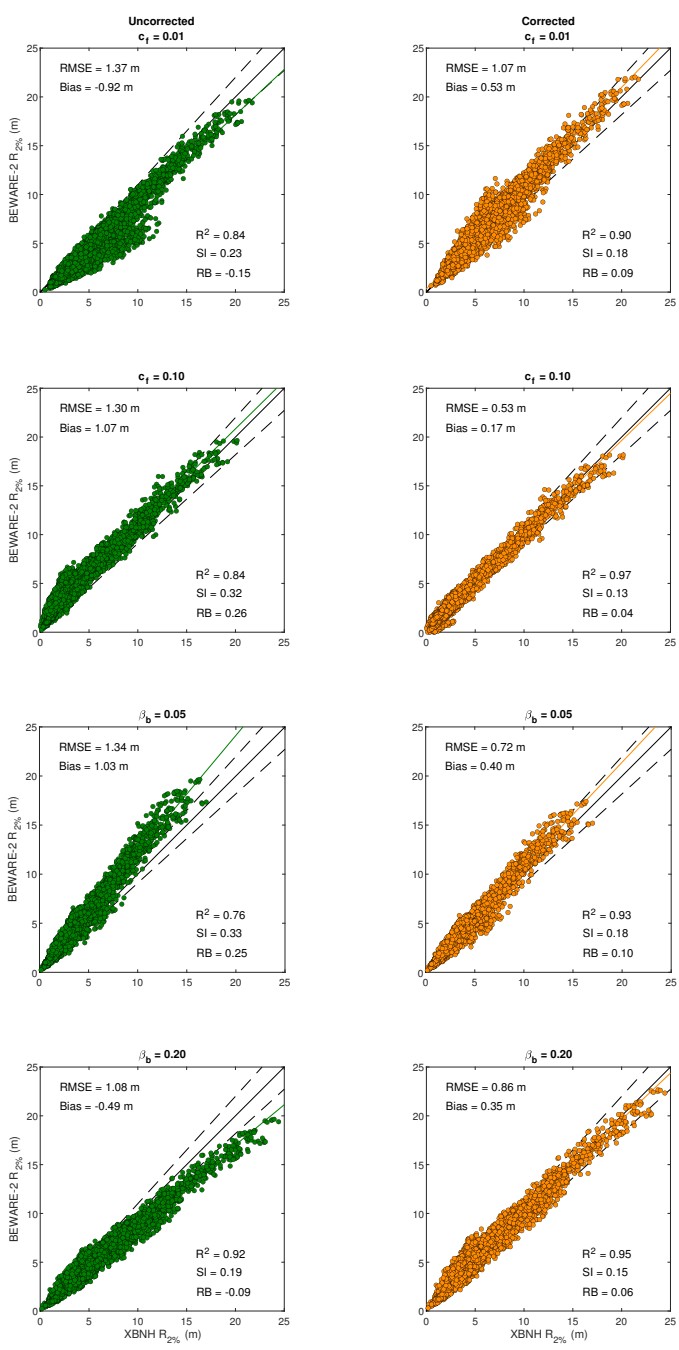

**Figure 9.** Wave runup simulated by XBNH (horizontal axis) and the expected value prediction of BEWARE-2 (vertical axis) for variations in the hydrodynamic roughness of the reef ($c_f$; upper two rows) and beach slope ($\beta_b$; lower two rows), without (left panels, green) and with (right panels, orange) application of Equations 4 and 8. Included here are the 1:1 relation (solid black line), 10% upper and lower deviation from 1:1 (dashed black lines) and the linear regression through the data (solid green and orange lines).




Application of Equations 4 and 8 improves model bias for most validation subsets of oceanic forcing, reef width, reef roughness, and beach slope (indicated by solid markers with values closer to zero in Figure 10). In particular, parametrizing the effect of reef roughness through Equation 4 most greatly improves model bias for moderate to high energy wave events ($H_s \geq 4$ m), moderate wave steepness ($0.02 \leq s_0 < 0.05$), low water levels (SWL $< 1$ m), and moderate to wide reef profiles ($W_{\text{reef}} \geq 500$ m). Conversely, parametrizing the effect of beach slope through Equation 8 most greatly improves model bias for low wave steepness ($s_0 < 0.02$), high water levels (SWL $\geq 2$ m), and narrow reef profiles ($W_{\text{reef}} < 500$ m).

Equations 4 and 8 do, however, systematically overpredict wave runup (also shown in Figure 9). In certain cases, this over-prediction may (in absolute sense) be greater than the underprediction of the uncorrected BEWARE-2 results. In particular, this is the case for the application of Equation 8 on steep beach profiles ($\beta_b = 0.20$), in combination with high energy wave events ($H_s \geq 8$ m), moderate to steep waves ($s_0 \geq 0.02$), low to moderate water levels (SWL $< 2$ m), or moderate to wide reef profiles ($W_{\text{reef}} \geq 500$ m). With respect to the application of Equation 4, deterioration in Bias is found only for the validation subset with narrow reef profiles ($W_{\text{reef}} < 500$ m), in combination with decreased reef roughness ($c_f = 0.01$). Note that application of Equations 4 and 8 improves model bias for all oceanic forcing conditions and reef widths in case of increased reef roughness ($c_f = 0.10$) and milder beach slopes ($\beta_b = 0.05$).


**Figure 10.** Bias in BEWARE-2 wave runup for simulations with increased ($\triangle$) or decreased ($\triangledown$) reef roughness ($c_f$; horizontal axis) and beach slope ($\beta_b$; vertical axis). Markers are shown for simulations without (green) and with (orange) the use of Equations 4 and 8. Solid green markers indicate that the result without the use of Equations 4 and 8 has the lowest absolute Bias value, whereas solid orange markers indicate lesser Bias with the use of Equations 4 and 8. Results are grouped by deep water wave height (top row), deep water wave steepness (second row), still water level (third row) and reef width (bottom row).



## 4 Discussion

BEWARE-2 was developed to provide a quick, accurate prediction of wave-driven total water levels (setup plus runup) across a wide range of extrinsic oceanographic forcing conditions (SWL, $H_{s,0}$ and $T_p$) and intrinsic coastal morphological parameters (reef bathymetry, $c_f$, and $\beta_b$). Over a range of 4 m in SWL, 10 m in $H_{s,0}$, and 16 s in $T_p$ across the 48 validation profiles

(spanning the range of reef morphologies in seven geographic regions), the meta-process modeling system produced results with an RMSE of 0.63 m and SI of 0.13, with 67% of all $R_{2\%}$ values falling within the 50% confidence interval prediction and 92% within the 90% confidence interval prediction of the full, process-based hydrodynamic XBNH model. Incorporating modifications to modeling system to account for variations in $c_f$ and $\beta_b$ allows systematic errors (RB) in BEWARE-2 predictions to be reduced by a factor of 1.5–6.5. Once having been trained, this relatively accurate solution is provided by the BEWARE-2

meta-process modeling system 4–5 orders of magnitude computationally faster (i.e., $\mathcal{O}(10^{-2})$ s per $R_{2\%}$ prediction) than the full, process-based hydrodynamic XBNH model ($\mathcal{O}(10^2$–$10^3)$ s per $R_{2\%}$ prediction) on a standard desktop computer.

### 4.1 Benefits and limitations of BEWARE-2 model

A key advantage of the BEWARE-2 meta-process modeling system is its ability to estimate uncertainties in the computed runup metrics. The uncertainty range is given by the range of weighted runup results obtained from the XBNH model simulations

for the matched RRPs and the nearest (upper and lower) oceanic forcing conditions (SWL, $H_{s,0}$, and $T_p$). This indication of uncertainty in the computed runup metric (i.e., $R_{2\%}$) would require multiple simulations and post-processing in a deterministic approach, which would come at great computational cost. However, in this implementation the meta-process modeling system will never be equal or better than the underlying process-based model.

The XBNH model is reasonably well validated, as described in the Introduction and Methods sections, but is not perfect. A

key limitation of BEWARE-2 is the one-dimensional nature of the underlying XBNH model simulations and thus the meta-process model dataset, which is unlikely to be very accurate in highly two-dimensional situations, such as in the presence of reef channels (e.g., Storlazzi et al., 2022). In addition, BEWARE-2 shows notably lower skill in reproducing XBNH runup values for very wide reef profiles (characteristic of barrier and extremely wide fringing reefs), likely due to the relatively low profile coverage (only 20 RRPs) and large number of possible permutations. Although this may be improved by including a

greater number of wide RRPs in the dataset, the real-world accuracy of the underlying XBNH model may likely be lower for very wide reef profiles than other reefs for which it has been validated, as one-dimensional XBNH models are unable to account for wind growth of waves or large-scale topographic refraction (Scott et al., 2020). Further extension of BEWARE-2 for these reef types is therefore currently constrained by the inherent uncertainty in applying XBNH to very wide reef profiles. A final key limitation of BEWARE-2 is the underlying reef profile dataset, which is heavily biased toward data from U.S.

(fringing) coral reef-lined coasts (due to consistent LiDAR data availability at the time of BEWARE-2 development) and may lack important other characteristic reef profiles such as barrier and platform reefs.





## 4.2 Potential applications of BEWARE-2

The BEWARE-2 meta-process modeling system is fast enough for large-scale application in EWS. As noted by Winter et al. (2020), a coastal flooding EWS would require four main modules: sea levels (including tides and non-tidal residuals), offshore
wave conditions, nearshore waves and water levels, and coastal flooding. BEWARE-2 can use the output of global or regional tide models (based on tide station observations and/or satellite altimetry, e.g., TPXO; Egbert and Erofeeva, 2002), global (e.g., HYCOM; Halliwell et al., 1998) or regional general circulation models, or a mix of these models, as SWL input. Offshore wave conditions ($H_{s,0}$ and $T_p$) for BEWARE-2 can be provided by operational global wave models, such as NOAA GEFS-Wave (utilizing the WaveWatch-III spectral wave model of Tolman, 2009), or region-specific alternatives. Using these oceanographic
inputs, BEWARE-2 provides runup, and, in the future, potentially nearshore waves, water levels and overtopping volumes that can be used to assess coastal flooding and thus form the core of an EWS for reef-lined coasts. The primary computational cost of developing an EWS based on BEWARE-2 is pre-processing; developing the BEWARE-2 database and matching of measured reef profiles to the RCP profiles. Once the desired real-world reef profiles have been assigned to the RRPs and the BEWARE-2 database has been linked to the operational water level and wave forecasts, the modeling system can provide a
statistical value (e.g., $R_{2\%}$) of runup and measures of its confidence within seconds.

BEWARE-2 can also provide a very rapid first approximation of runup and flooding potential for use in flood risk analyses. In such analyses, the influence reef health (here expressed in terms of reef roughness following Quataert et al., 2015, and Norris et al., 2023) on runup, and hence flood risk, can be calculated using the BEWARE-2 reef roughness parametrization. Due to the speed of the system, BEWARE-2 can be used simulate 100s years of runup on a range of profiles with varying
reef health to allow climate hindcast and future climate scenario analysis of wave runup. This could help identify coastlines at particular risk of higher and/or more frequent wave runup in the future. Alternatively, BEWARE-2 results can be used to identify relevant nearshore wave and water level conditions in long time series for simulation in other, more computationally expensive, numerical models (e.g., Masselink et al., 2020, 2021).

## 4.3 Next steps

The BEWARE-2 meta-process modeling system and the underlying process-based hydrodynamic XBNH model ultimately require field validation of both waves and water levels over reefs, and the resulting wave-driven runup on the coast. Such measurements, especially runup, are scarce, but are slowly increasing with the deployment of coastal imaging systems, such as Argus (Holman and Stanley, 2007), in tropical, reef-lined locations. Further testing should also include profiles from non-U.S. reefs that are not included in the dataset used to develop the RRPs, prior to application in other regions.
One of the great aspects of the BEWARE-2 database is its modular nature. The range of oceanographic forcing conditions can be expanded (lower or higher water level and/or wave heights and periods) or refined by additional XBNH simulations. Non-U.S. reef profiles that are morphologically distinct from the current dataset can be included through the addition of new RRPs. Similarly, if one wanted to investigate the role of coral reef restoration in the reef profiles, new RRPs that include the effects of restoration (changes in height and/or bed roughness) could be simulated in XBNH (e.g., Roelvink et al., 2021)



and added to the database. Lastly, one could utilize information in the BEWARE-2 dataset to extend prediction variables with output such as overtopping volumes for use in coastal flood modeling studies.

## 5    Conclusions

A surrogate, meta-process model (BEWARE-2) of the process-based hydrodynamic model XBeach Non-Hydrostatic+ (XBNH; de Ridder et al., 2021) has been developed to estimate wave runup on morphologically diverse reefs. BEWARE-2 builds on work by Pearson et al. (2017) and Scott et al. (2020) to account for a broad range of oceanic forcing conditions (water levels, wave height and period), diverse morphologies of naturally occurring coral reef profiles, and variations in reef roughness and beach slope.

The BEWARE-2 meta-process model accurately reproduces runup (2% exceedance runup level; $R_{2\%}$) simulated by the process-based XBNH model on 48 real-world profiles drawn from seven geographic regions. BEWARE-2 was tested for 100 oceanic forcing conditions (combinations of offshore water level, wave height and period), with simulated runup ranging from 0.17 to 20.9 m. BEWARE-2 was shown to have a high overall coefficient of determination ($R^2 = 0.96$) and an overall root-mean square error of 0.63 m (scatter index: 0.13) and bias of 0.26 m (relative bias: 0.05). Model skill differed little for variations in oceanic forcing conditions, but was substantially greater for profiles with reef widths less than 1.5 km ($R^2 = 0.98$, scatter index = 0.10, relative bias = 0.05) than for wider reef profiles ($R^2 = 0.72$, scatter index = 0.25, relative bias = 0.08). Parametric extensions to BEWARE-2 to account for variations in reef roughness and beach slope were shown to reduce systematic errors (relative bias) in BEWARE-2 predictions by a factor of 1.5–6.5 for relatively coarse or smooth reefs, and mild or steep beach slopes.

The simulation of wave runup using BEWARE-2 is 4–5 orders of magnitude faster ($\mathcal{O}(10^{-2})$ s per $R_{2\%}$ prediction) than simulating runup using XBNH ($\mathcal{O}(10^2$–$10^3)$ s per $R_{2\%}$ prediction). In addition, the framework of the BEWARE-2 meta-process model provides a probability distribution of wave runup that can be used to determine the expected value of $R_{2\%}$, alongside confidence bands around this prediction. Key limitations of BEWARE-2 are related to the one-dimensional nature of underlying XBNH model simulations, both for application in highly two-dimensional situations and on very wide reef profiles, as well as the underlying reef profile dataset, which is heavily biased toward data from U.S. coral reef-lined coasts.

The accuracy and speed of the BEWARE-2 meta-process model suggest that it may be a useful tool for early warning systems and current and future coastal flood risk analysis.

*Data availability.*    The BEWARE-2 database presented in this paper is available as a NetCDF (*.nc) file, hosted at the following location: https://doi.org/10.5066/XXXXXXXX [to be added at publication].





## Appendix A: Accounting for variations in reef roughness and beach slope

### A1 Calibration dataset

In a similar approach to the creation of the BEWARE-2 training dataset (Section 2.1.3), a separate and substantially smaller dataset was created for the purpose of developing and calibrating BEWARE-2 wave runup correction factors for smoother and rougher reefs, and steeper and milder beach slopes. To this end, XBNH simulations were carried out on 31 morphologically diverse reef profiles (Figure A1) that were selected through agglomerative hierarchical clustering of the 530 normal reef iRRPs and that represent a subset of the BEWARE-2 RRPs. XBNH simulations were forced using a subset of the hydrodynamic

boundary conditions included in the BEWARE-2 training dataset (Table A1). XBNH simulations for all calibration reef profiles and hydraulic boundary conditions were run with the reference reef roughness ($c_{f,\mathrm{ref}} = 0.05$) and beach slope ($\beta_{b,\mathrm{ref}} = 0.10$), as well as for higher ($c_f = 0.10$) or lower ($c_f = 0.01$) reef roughness, or steeper ($\beta_b = 0.20$) or milder ($\beta_b = 0.05$) beach slope (i.e., five $c_f$–$\beta_b$ combinations in total). Other XBNH model parameters were set equal to those used for the BEWARE-2 training dataset (Section 2.1.2). XBNH $R_{2\%}$ values for the calibration dataset were calculated as described in Section 2.1.3.

| Forcing parameter | Value |
|---|---|
| SWL (m + MSL) | 0, 1, 2, 4 |
| $H_{s,0}$ (m) | 1, 3, 5, 7, 9, 11 |
| $T_p$ (s) | 8*, 12, 16, 20 |

**Table A1.** Overview of hydrodynamic forcing conditions used for the calibration of reef roughness and beach slope effects on wave runup. *Wave height combinations for $T_p = 8$ s are constrained by maximum deep water wave steepness.

### A2   Parameterization of reef roughness effect

The effect of increased or reduced roughness of the reef relative to $c_{f,\mathrm{ref}}$ on wave runup is not included in the underlying dataset of BEWARE-2. Instead, this effect is parametrized in the model through estimation of the relative increase or decrease in wave height at the toe of the beach due to roughness effects, under the assumption that changes in wave runup will be proportional to changes in the wave height reaching the beach:

$$\Delta R_{2\%} \propto \Delta H_{\mathrm{beach\ toe}} \tag{A1}$$

To estimate the contribution of reef roughness effects on the wave height at the beach toe (defined as the landward edge of the reef located at a depth of 0.5 m, see Section 2.1.2), we assume a stationary wave energy balance on the reef profile:

$$\frac{\partial c_g(x)E(x)}{\partial x} = -D_f(x) \tag{A2}$$

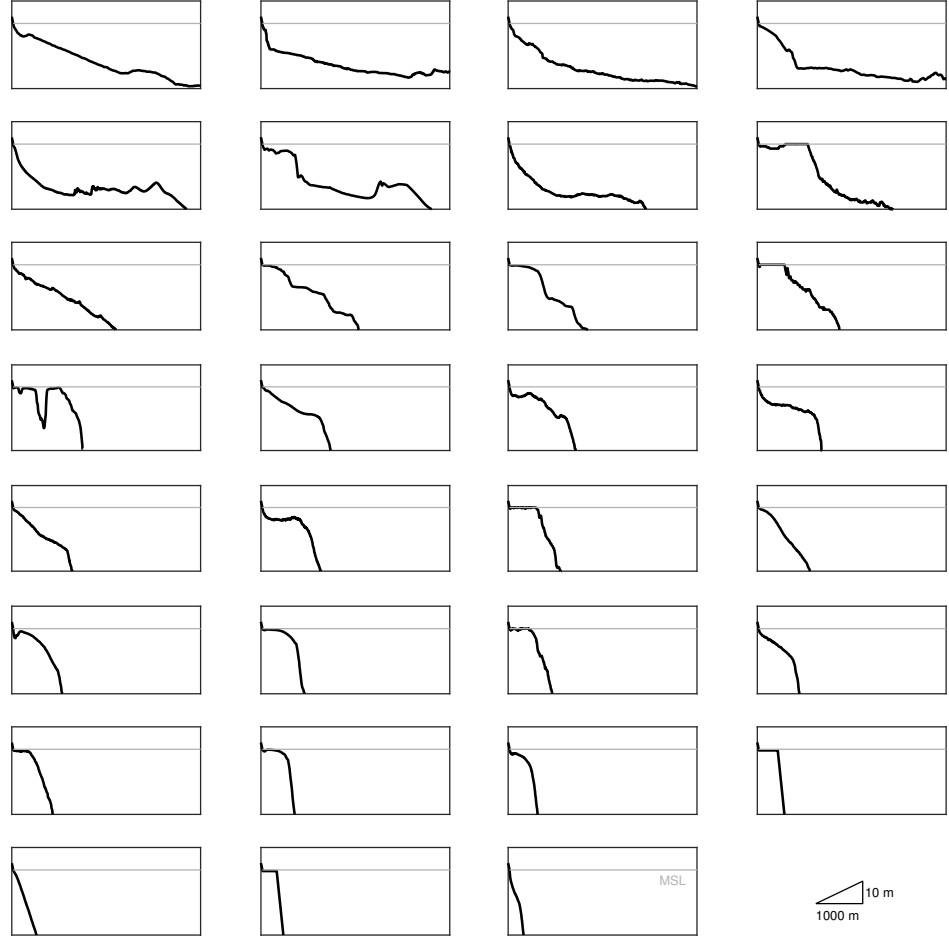

**Figure A1.** Cross-shore profiles of 31 morphologically diverse RRP reef transects derived by agglomerative hierarchical clustering used to calibrate the parameterization of bed roughness and beach slope effects on wave runup.

in which $E$ is the wave energy, $c_g$ is the wave group velocity, $x$ is the cross-shore coordinate, and $D_f(x) = \frac{2}{3\pi} c_f \rho u_{\mathrm{orb}}^3(x)$ is the wave dissipation term due to reef roughness effects, $\rho$ is the density of water and $u_{\mathrm{orb}}$ is the near-bed wave orbital velocity. Note that dissipation due to depth-induced breaking is intentionally not included in Equation A2, but is instead imposed as a boundary condition to the solution in Equation A8.

Through linearisation of the wave group velocity in Equation A2, the integrated wave energy loss across the reef due to reef roughness effects only ($\Delta E_r$) is approximated by:

$$\Delta E_r \approx - \int_{L_{c_f}} \frac{D_f(x)}{c_g(x)} dx \tag{A3}$$




in which $L_{c_f}$ is the cross-shore extent of the model domain that has a hydrodynamic roughness value equal to $c_{f,\text{ref}}$, i.e., the cross-shore extent of the coral reef.

Given $H = \sqrt{\frac{8E}{\rho g}}$, the wave height loss across the reef due to reef roughness effects only ($\Delta H_r$) is proportional to:

$$\Delta H_r \propto -\sqrt{\frac{c_f}{g}}\,\Gamma_{\text{reef}} \tag{A4}$$

where $\Gamma_{\text{reef}}$ is the integrated wave dissipation estimate defined in Equation 7.

The assumed proportionality of Equation A1 is combined with the estimated difference in wave height loss across the reef due to varying reef roughness, normalized by the offshore significant wave height ($H_{s,0}$), to derive a proportionality for relative variations in wave runup due to reef roughness effects:

$$\frac{R_{2\%,c_f} - R_{2\%,c_{f,\text{ref}}}}{R_{2\%,c_{f,\text{ref}}}} \propto \frac{\sqrt{\frac{1}{g}}\sqrt{\left|\frac{c_f}{c_{f,\text{ref}}} - 1\right|}\sqrt{\int_{L_{c_f}} \frac{u_{\text{orb}}(x)^3}{c_g(x)}dx}}{H_{s,0}} = \frac{\sqrt{\frac{1}{g}}\gamma_r \Gamma_{\text{reef}}}{H_{s,0}} \tag{A5}$$

where $R_{2\%,c_{f,ref}}$ is the wave runup computed for the reference reef roughness value ($c_{f,\text{ref}} = 0.05$) using the underlying XBNH dataset, $R_{2\%,c_f}$ is the estimated wave runup for reef profiles with a higher or lower roughness ($c_f$) than the reference roughness, and $\gamma_r$ is the relative reef roughness coefficient defined in Equation 6.

Solving the right hand term in Equation A5 requires an estimate of the local near-bed wave orbital velocity, which is computed as:

$$u_{\text{orb}}(x) = \frac{\pi H_{\text{rms}}(x)}{T_p}\frac{1}{\sinh\left(k_p(x)h_0(x)\right)} \tag{A6}$$

where $H_{\text{rms}}$ is the root-mean-square wave height, $k_p$ is the wave number based on $T_p$ and $h_0$ is the local water depth estimated using the cross-shore profile elevation and the offshore still water level (SWL).

The root-mean-square wave height at the offshore boundary of the reef profile ($H_{\text{rms},0}$) is estimated as:

$$H_{\text{rms},0} = \frac{H_{s,0}}{\sqrt{2}} \tag{A7}$$

Wave height transformation across the reef profile is estimated using a forward-marching scheme, starting at the offshore boundary of the profile. In this, the computed wave height includes wave shoaling and depth-induced breaking processes, but intentionally neglects dissipation due to hydrodynamic bed roughness:

$$H_{\text{rms}}(x(i)) = \min\left(H_{\text{rms}}(x(i-1))\sqrt{\frac{c_{g,p}(x(i-1))}{c_{g,p}(x(i))}}, \gamma_b h_0(x(i))\right) \tag{A8}$$


where $i$ is the numerical grid cell counter, starting at $i = 1$ at the offshore boundary and increasing in cross-shore direction,
$c_{g,p}$ is the wave group velocity estimated using $T_p$, and $\gamma_b = 0.78$ is the wave breaker index.

Equation A8 allows a first order estimate of wave heights across the reef in the absence of reef roughness effects. Although this method neglects important physical processes (e.g., wave set-up, infragravity wave generation), we find that this first order estimate provides sufficient information with which to derive proportionality constants for Equation A5 and subsequently improve predictions of wave runup on reef profiles with greater and lesser roughness (Section 3.3).

Linear proportionality coefficients ($\alpha_r$) for Equation 5 (and Equation A5) are found separately for relatively smooth ($c_f = 0.01$) and rough ($c_f = 0.10$) reef profiles by computing the left-hand and right-hand terms of Equation A5 for the reef roughness calibration simulations described in A1 and fitting linear least-square error relations to the smooth and rough reef datasets (Figure A2). In this manner, we find proportionality coefficients $\alpha_{r,\text{smooth}} = 1.16$ for relatively smooth reefs and $\alpha_{r,\text{rough}} = -0.65$ for relatively rough reefs, with considerable scatter around the linear fit that is likely due to the extensive simplification
of the true hydrodynamics on reef profiles.

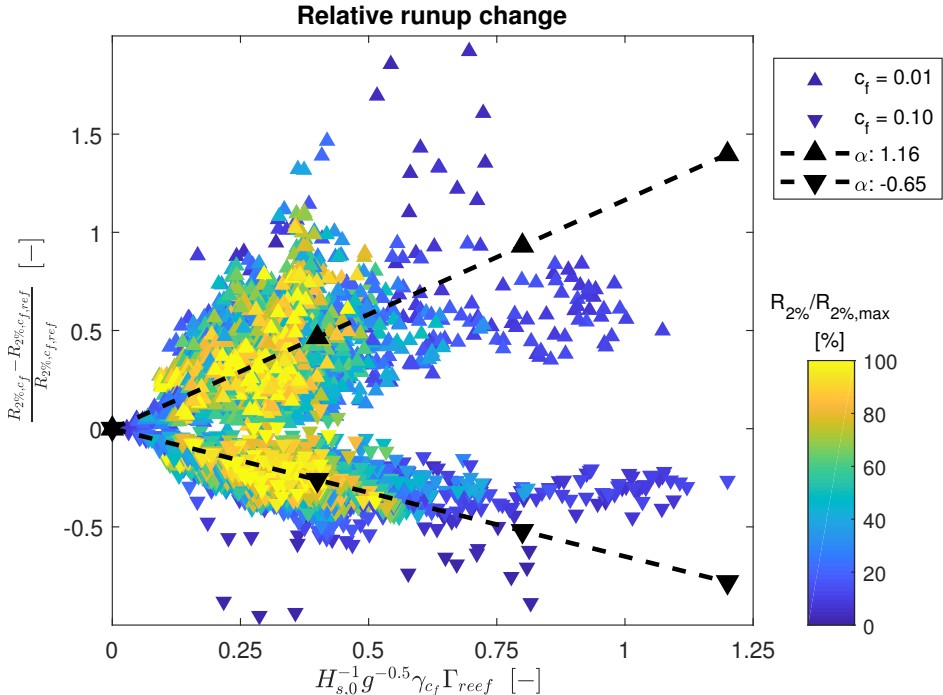

**Figure A2.** Variation in relative wave runup as a function of the right-hand term of Equation A5. Linear proportionality constants $\alpha_r$ found for reef profiles with lower ( $c_f = 0.01$; ▲) and higher ($c_f = 0.10$; ▼) reef roughness are shown as dashed lines. Computed relative runup values are coloured by the computed wave runup relative to the maximum runup in the calibration dataset.

Despite the high degree of simplification present in this methodology, application of the calibrated $\alpha_r$ coefficients to the conditions included in the calibration dataset (A1) greatly improves skill metrics of wave runup (Figure A3). In particular,





application of the calibrated coefficients greatly reduces the bias of the runup predictions (i.e., improvement of RB from -0.25 to 0.00 for smoother reefs, and from 0.30 to 0.02 for rougher reefs), as would be expected from the linear fit methodology.

Importantly, as shown in Section 3.3, application of the calibrated $\alpha_r$ coefficients to the validation dataset also substantially reduces model bias and improves the wave runup skill metrics of BEWARE-2.

The two derived values of $\alpha_r$ appear to scale approximately proportionally with the square root of the change in the reef roughness coefficient ($\alpha_{r,\text{smooth}} \approx 0.5\sqrt{\frac{0.05}{0.01}}$; $\alpha_{r,\text{rough}} \approx -0.5\sqrt{\frac{0.10}{0.05}}$), with a change in sign for rougher reefs relative to the reference reef roughness. However, the use of inter- or extrapolated values of $\alpha_r$ for reefs with other representative roughness

values is not recommended without further investigation of this relation. Instead, wave runup values computed in BEWARE-2 using $\alpha_{r,\text{smooth}}$ and $\alpha_{r,\text{rough}}$ can be used to illustrate likely upper and lower bounds of wave runup for natural reefs in poor or good health.

### A3  Parameterization of beach slope variation effect

Similarly to reef roughness variations, the effect of steeper or milder beach slopes relative to $\beta_{b,\text{ref}}$ on wave runup is not

included in the underlying dataset of BEWARE-2. Instead, this effect is parametrized in the model through estimation of the contribution of the beach slope to the incident-band swash and swash-zone setup.

Following Stockdon et al. (2006), we can describe the components of wave runup as:

$$R_{2\%} = c_R\left(\overline{\eta_{\text{wl}}} + \frac{\sqrt{S_{\text{IG}}^2 + S_{\text{inc}}^2}}{2}\right) \tag{A9}$$

where $\overline{\eta_{\text{wl}}}$ is the time-averaged vertical position of the waterline observed at the shore, $S_{\text{IG}}$ and $S_{\text{inc}}$ are the infragravity and

incident-band swash, respectively, and $c_R$ is a regression coefficient found to have a value of 1.1 in the data of Stockdon et al. (2006).

Following Stockdon et al. (2006), we assume that variation in beach slope (in this case $\beta_b = 0.05$–$0.20$) will not affect $S_{\text{IG}}$. Furthermore, we assume that the contribution of the surf-zone setup ($\overline{\eta_{\text{surf}}}$) to the setup observed at the shoreline ($\overline{\eta_{\text{wl}}}$) is independent of the beach slope (but instead dependent upon the shoreface and reef profile) and therefore already correct in the

underlying training dataset for BEWARE-2. It remains, however, that a beach slope correction factor ($\alpha_b$) is likely required for the incident-band swash ($S_{\text{inc}}$) and setup occurring within the swash zone ($\overline{\eta_{\text{swash}}} = \overline{\eta_{\text{wl}}} - \overline{\eta_{\text{surf}}}$) to improve wave runup predictions in BEWARE-2 for beach slopes different to $\beta_{b,\text{ref}}$.

To determine the value of $\alpha_b$ we first empirically determine the values of $R_{2\%}$, $\overline{\eta_{\text{wl}}}$, $S_{\text{IG}}$, and $S_{\text{inc}}$ from the time series of the waterline elevation (i.e., time-varying swash signal) output by XBNH in all calibration simulations with the reference beach

slope ($\beta_{b,\text{ref}} = 0.10$). In this, we compute $R_{2\%}$ as the empirical 2%-exceedance height of swash peaks above still water level, and $S_{\text{IG}}$ and $S_{\text{inc}}$ from the variance density spectrum of the simulated waterline elevation time series (see e.g., Stockdon et al. 2006; de Beer et al. 2021) using frequency integration limits of $\frac{1}{20}f_p \leq f < \frac{1}{2}f_p$ for the computation of $S_{\text{IG}}$ and $\frac{1}{2}f_p \leq f < 3f_p$ for the computation of $S_{\text{inc}}$, where $f$ is frequency and $f_p = \frac{1}{T_p}$ is the offshore wave spectral peak frequency. We derive $\overline{\eta_{\text{surf}}}$ from the mean water level at the nearest XBNH model output point to the toe of the beach, which is located at MSL - 0.5 m and

**Figure A3.** Wave runup predicted by BEWARE-2, without (left panels) and with (right panels) use of $\alpha_r$, relative to simulated by XBNH for smoother ($c_f = 0.01$; top panels) and rougher ($c_f = 0.10$; bottom panels) reefs than the reference reef roughness ($c_{f,\mathrm{ref}} = 0.05$) in the calibration dataset. Figures include the 1:1 relation (solid black line), 10% upper and lower deviation from 1:1 (dashed black lines) and the linear regression through the data (solid green and orange lines).


$\overline{\eta_{\text{swash}}}$ from the time-averaged waterline elevation. We subsequently calculate the value of $c_R$ separately for every simulation as

$c_R = \dfrac{R_{2\%}}{\overline{\eta_{\text{wl}}} + \frac{\sqrt{S_{\text{IG}}^2 + S_{\text{inc}}^2}}{2}}$, thereby ensuring that application of Equation A9 with simulation-specific values of $c_R$ provides an exact

representation of the waterline time series-derived $R_{2\%}$-value. The values of $c_R$ computed in this manner range between 0.92

and 2.04 (median value 1.16).

Given the computed values of $c_R$, $\overline{\eta_{\text{surf}}}$, $\overline{\eta_{\text{swash}}}$, $S_{\text{IG}}$ and $S_{\text{inc}}$ for all calibration simulations with the reference beach slope

$\beta_{b,\text{ref}}$, we can estimate the wave runup for other beach slopes ($R_{2\%}^{m,b}$) through inclusion of $\alpha_b$ in Equation A9 as follows:

$$R_{2\%}^{m,b} = c_R \left( \overline{\eta_{\text{surf}}} + \alpha_b \overline{\eta_{\text{swash}}} + \frac{\sqrt{S_{\text{IG}}^2 + (\alpha_b S_{\text{inc}})^2}}{2} \right) \tag{A10}$$

To test the validity of Equation A10, values of $R_{2\%}^{m,b}$ are compared to runup values derived empirically from XBNH simulations for the same RRPs and forcing conditions, but with beach slope set at $\beta_b = 0.05$ and $\beta_b = 0.20$ (see Section A1).

As Stockdon et al. (2006) found both $\overline{\eta_{\text{wl}}}$ and $S_{\text{inc}}$ to be proportional to $\beta_b$, the required beach slope correction factor $\alpha_b$

would be expected to be proportional to the change in beach slope relative to the reference beach slope (i.e., $\alpha_b = \frac{\beta_b}{\beta_{b,\text{ref}}}$).

However, we find that if a linear relation between $\beta_b$ and $\alpha_b$ is used, Equation A10 greatly overestimates the effect of the beach slope on wave runup relative to that simulated by XBNH, showing underestimation of wave runup relative to XBNH for milder beach slopes (Figure A4, top right panel) and overestimation for steeper beach slopes (Figure A4, bottom right panel). In quantitative sense, application of Equation A10 and a linear relation between $\beta_b$ and $\alpha_b$ even reduces the predictive skill of

the model for steep beach slopes ($\beta_b = 0.20$) relative to applying no beach slope correction at all (Figure A4, bottom panels).

Substantial improvement in model predictive skill is found, however, if a non-linear relation is assumed between $\beta_b$ and $\alpha_b$ in the form of $\alpha_b = (\frac{\beta_b}{\beta_{b,\text{ref}}})^{\kappa_b}$. Through minimization of the combined relative bias (RB) of the runup prediction for both milder and steeper beach slopes, optimal values of $\kappa_b$ were found in the range 0.35–0.38 (difference in combined RB across range less than 0.5 percentage points), with a value $\kappa_b = \frac{1}{e} \approx 0.37$ selected for application in the model (Equation 10).

Application of Equation 8 using $\kappa_b = \frac{1}{e}$ substantially improves the skill of BEWARE-2 in reproducing XBNH wave runup on milder and steeper beaches in the calibration dataset, see Figure A5. The non-linear nature of Equation 10 is, however, not fully understood and extrapolation for beach slopes outside the range of the calibration simulations (i.e., $\beta_b = 0.05$–0.20) is therefore not recommended.


**Figure A4.** Wave runup predicted by BEWARE-2 relative to simulated by XBNH for milder (top panels) and steeper (bottom panels) beach slopes than the reference beach slope ($\beta_{b,\text{ref}} = 0.10$) in the calibration dataset. Predictions are without beach slope correction (left panels) and with a correction factor $\alpha_b$ linearly proportional to $\beta_b$ ($\kappa_b = 1$). Note the detrimental effect of the linear beach slope correction on the model skill in the bottom-right panel.



**Figure A5.** Wave runup predicted by BEWARE-2 relative to simulated by XBNH for milder (top panels) and steeper (bottom panels) beach slopes than the reference beach slope ($\beta_{b,\mathrm{ref}} = 0.10$) in the calibration dataset using $\kappa_b = \frac{1}{e}$.



*Author contributions.*  RM, CS, SP, and JA formulated the overall research aims and methodology; RM, FR, SP, JA, and RdG developed the
BEWARE-2 software; CS provided data and supervised BEWARE-2 development; JA and FR extended the probabilistic profile matching
methodology; RM developed the reef roughness and beach slope correction factors; FR and RdG prepared the training and validation data;
RM, CS, and FR prepared the draft manuscript; SP and JA reviewed and edited the manuscript.

*Competing interests.*  The authors declare that they have no conflict of interest.

*Disclaimer.*  Any use of trade, firm, or product names is for descriptive purposes only and does not imply endorsement by the US Government.

*Acknowledgements.*  This research was funded by the U.S. Geological Survey's Coastal and Marine Hazards and Resources Program and
Deltares' Strategic Research in the Natural Hazards Program (11209194). We would like to thank Dr. M. Palmsten and Dr. B. Tsai for their
suggestions and constructive review prior to submission to this journal.




| | $N_{match}$ (-) | $p_{max}$ (%) | RMSE (m) | Bias (m) | SI (-) | RB (-) | $R^2$ (-) |
|---|---|---|---|---|---|---|---|
| AmSamoa: 5% | 4 | 80 | 0.78 | -0.06 | 0.08 | -0.01 | 0.98 |
| AmSamoa: 25% | 6 | 35 | 0.44 | 0.14 | 0.05 | 0.02 | 0.99 |
| AmSamoa: 50% | 10 | 27 | 0.51 | 0.31 | 0.08 | 0.05 | 0.96 |
| AmSamoa: 75% | 10 | 25 | 0.75 | 0.66 | 0.16 | 0.14 | 0.83 |
| AmSamoa: 95% | 4 | 60 | 0.59 | 0.51 | 0.16 | 0.14 | 0.73 |
| AmSamoa W | 6 | 91 | 1.17 | 1.01 | 0.35 | 0.30 | 0.01 |
| Florida: 5% | 7 | 58 | 0.41 | -0.17 | 0.09 | -0.04 | 0.93 |
| Florida: 25% | 4 | 68 | 0.31 | 0.15 | 0.08 | 0.04 | 0.95 |
| Florida: 50% | 4 | 48 | 0.30 | 0.04 | 0.08 | 0.01 | 0.95 |
| Florida: 75% | 4 | 87 | 0.38 | 0.26 | 0.13 | 0.09 | 0.87 |
| Florida: 95% | 4 | 92 | 0.37 | 0.30 | 0.15 | 0.12 | 0.87 |
| Florida W: 25% | 5 | 87 | 0.34 | 0.21 | 0.11 | 0.07 | 0.90 |
| Florida W: 75% | 3 | 70 | 0.39 | -0.20 | 0.24 | -0.12 | 0.75 |
| Guam: 5% | 6 | 49 | 0.51 | -0.00 | 0.06 | -0.00 | 0.99 |
| Guam: 25% | 10 | 40 | 0.61 | 0.44 | 0.09 | 0.07 | 0.95 |
| Guam: 50% | 10 | 36 | 0.81 | 0.72 | 0.16 | 0.14 | 0.82 |
| Guam: 75% | 7 | 38 | 0.54 | 0.46 | 0.16 | 0.14 | 0.79 |
| Guam: 95% | 4 | 56 | 0.43 | 0.29 | 0.17 | 0.11 | 0.80 |
| Guam W: 25% | 3 | 98 | 0.43 | 0.13 | 0.18 | 0.05 | 0.79 |
| Guam W: 75% | 4 | 53 | 0.25 | -0.18 | 0.12 | -0.08 | 0.89 |
| Hawaii: 5% | 4 | 76 | 0.63 | 0.06 | 0.06 | 0.01 | 0.98 |
| Hawaii: 25% | 9 | 50 | 0.66 | 0.42 | 0.09 | 0.06 | 0.96 |
| Hawaii: 50% | 8 | 42 | 0.46 | -0.18 | 0.09 | -0.03 | 0.95 |
| Hawaii: 75% | 4 | 64 | 0.44 | 0.35 | 0.12 | 0.10 | 0.89 |
| Hawaii: 95% | 4 | 85 | 0.61 | 0.52 | 0.21 | 0.18 | 0.64 |
| Hawaii W: 25% | 7 | 56 | 0.33 | -0.17 | 0.10 | -0.05 | 0.92 |
| Hawaii W: 75% | 4 | 85 | 1.16 | 1.02 | 0.43 | 0.37 | -0.53 |
| PuertoRico: 5% | 8 | 54 | 0.68 | 0.30 | 0.09 | 0.04 | 0.96 |
| PuertoRico: 25% | 10 | 37 | 0.63 | 0.52 | 0.11 | 0.09 | 0.93 |
| PuertoRico: 50% | 7 | 43 | 0.93 | 0.75 | 0.18 | 0.14 | 0.76 |
| PuertoRico: 75% | 7 | 29 | 0.44 | 0.32 | 0.11 | 0.08 | 0.88 |
| PuertoRico: 95% | 4 | 87 | 0.47 | 0.39 | 0.15 | 0.13 | 0.84 |
| PuertoRico W: 25% | 5 | 79 | 1.17 | -0.81 | 0.23 | -0.16 | 0.70 |
| PuertoRico W: 75% | 4 | 80 | 0.32 | -0.10 | 0.08 | -0.03 | 0.95 |
| SaipanTinian: 5% | 4 | 83 | 0.78 | -0.05 | 0.08 | -0.01 | 0.98 |
| SaipanTinian: 25% | 7 | 38 | 0.47 | -0.02 | 0.05 | -0.00 | 0.99 |
| SaipanTinian: 50% | 10 | 46 | 0.80 | 0.62 | 0.12 | 0.09 | 0.92 |
| SaipanTinian: 75% | 10 | 30 | 0.73 | 0.61 | 0.14 | 0.12 | 0.89 |
| SaipanTinian: 95% | 4 | 93 | 0.19 | 0.06 | 0.08 | 0.02 | 0.97 |
| SaipanTinian W: 25% | 4 | 94 | 0.51 | -0.10 | 0.20 | -0.04 | 0.78 |
| SaipanTinian W: 75% | 4 | 74 | 1.16 | 1.00 | 0.39 | 0.34 | -0.25 |
| USVI: 5% | 6 | 59 | 0.56 | 0.03 | 0.06 | 0.00 | 0.99 |
| USVI: 25% | 8 | 42 | 0.42 | -0.08 | 0.06 | -0.01 | 0.98 |
| USVI: 50% | 6 | 64 | 0.42 | 0.20 | 0.07 | 0.04 | 0.97 |
| USVI: 75% | 4 | 89 | 0.41 | 0.23 | 0.09 | 0.05 | 0.94 |
| USVI: 95% | 6 | 87 | 0.24 | 0.12 | 0.08 | 0.04 | 0.95 |
| USVI W: 25% | 4 | 96 | 0.53 | 0.24 | 0.14 | 0.06 | 0.88 |
| USVI W: 75% | 9 | 50 | 0.44 | 0.38 | 0.17 | 0.15 | 0.74 |

**Table A1.** Overview of profile matching properties and model skill values for the 48 validation profiles. Matching properties include the number of probabilistically matched RRPs ($N_{match}$) and the maximum matching probability ($p_{max}$), and skill values include the root-mean-square error (RMSE), model bias (Bias), scatter index (SI), relative bias (RB), and coefficient of determination ($R^2$) of the BEWARE-2 wave runup prediction.



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
