# Peer review of "Rapid simulation of wave runup on morphologically diverse, reef-lined coasts with the BEWARE-2 meta-process model"

_Natural Hazards and Earth System Sciences, 2024_

## Author Comment (AC1)

**Response to Referee #1 (Ron Hoeke) comments on**

*Rapid simulation of wave runup on morphologically diverse, reef-lined coasts with the BEWARE-2 meta-process model*

*Robert McCall, Curt Storlazzi, Floortje Roelvink, Stuart Pearson, Roel de Goede, and José Antolínez*

We would like to thank Dr Hoeke for his suggestions and constructive comments on this manuscript. We have attempted to address the points made by the referee in the updated manuscript and/or have provided our rebuttal below. In the following, the referee's comments are given in **black** font and our response in **blue** font.

We have been advised that we are not able to upload the updated manuscript to the discussion portal at this time. In lieu of providing the referee with an updated copy of the manuscript, we have therefore included screenshots of changes to the manuscript, where appropriate, at the end of this document.

1. Overall: In my view, this paper is what it says it is, i.e.: "a useful tool for early warning systems and current and future coastal flood risk analysis" for a broad range of fringing-reef morphologies. Given the high uncertainties of coastal flood risk and lack of available EWS across much of the world's vulnerable reef-lined coasts, it makes it very worth reporting.

   We thank the referee for his support of the topic addressed in this manuscript.

   However, the paper could be substantially improved in several ways. Among them:

2. The authors should be more up-front and clearer about what is different between this paper and the earlier BEWARE paper (Pearson et al., 2017), which many readers may already be familiar with. Besides the addition of (a lot) of new (real-world) profiles and related training data for the surrogate model, why has the Bayesian approach apparently been abandoned? Or are you just not calling the training steps "Bayesian" anymore?

   This suggestion was also provided by Referee #2. We have attempted to clarify the differences between the original BEWARE model (Pearson et al., 2017) and the BEWARE-2 model in the introduction section (e.g., lines 74–78, see also screenshot in Figure 3 at the end of this document). The most important difference that we aim to highlight is the move from parametric reef profile shapes used in BEWARE (and other metamodels referred to later by the referee), which are too simplified to well describe the natural extreme bathymetric variability of coral reefs (Scott et al., 2020), towards the 195 representative reef profiles (RRPs) used in BEWARE-2, which encompass a far greater variability in reef geometries seen across the globe. The Bayesian approach to estimate wave runup has indeed been replaced in BEWARE-2 by probabilistic matching of target profiles to the RRPs and weighted nearest neighbor probability matching of target oceanic forcing conditions to database conditions. This modification in probabilistic approach is now explicitly stated in Section 2.2 (lines 226–229, see also screenshot in Figure 6 at the end of this document).

3. Related - most of the complex logic appears to be used for matching the target reef profile to the representative reef profiles (RRPs, which were developed primarily in an earlier work); comparatively simple inverse distance weighted interpolation of the full-fidelity (XB-NH) model "training data" is then seems to be used to estimate target profile and target conditions (albeit with some interesting heuristic relationships used to post-hoc estimate effects of bed friction and beach slope). That seems (in my experience anyway) a different approach compared to most coastal hybrid/meta-models, which seek to emulate the dynamics themselves over a given morphology (e.g. Zornoza-Aguado, et al 2024). Would it not be easier (in the modern age) supply all training data (including the reef profiles themselves) to some kind of conditioned neural network (NN), either a simple one, such as the RBF approach used by Rueda et al 2019

and others, or a deep NN, or explore any of the rapidly evolving more complex black-box ML approaches? Maybe you don't need that level of complexity due to the profile 1-D nature of the problem and a more first-principle morphological approach is better? I think the explaining the rational used here and how it diverges (or doesn't) from other contemporary meta/hybrid modelling approaches for coastal extremes would greatly improve the paper.

The referee makes an interesting point here regarding the need for (or usefulness of) more complex machine learning (ML) methods with which to train the BEWARE-2 model. This is a question that we had also considered during the development and training. We found, in line with the referee's statement, and as explored in earlier work by Scott et al. (2020), that most of the complexity in the ML methodology is required to simplify the multidimensional geometric parameter space (in this case through probabilistic matching to RRPs). In contrast, training across the gridded and (relatively) small dimensional space of oceanic forcing conditions appeared to be easily achieved using a relatively simple ML method (essentially a weighted nearest neighbor approach).

In general sense, we expect inverse distance weighted methods, such as the method we apply in BEWARE-2, to be comparable in performance to Global Basis Function-type methods (such as RBF, splines, etc.), as long as the data we are using are gridded, as is the case for the oceanic forcing conditions. If the oceanic forcing condition training data had been scattered, we would expect Global Basis Function-type methods to outperform our more simple weighted distance approach.

Despite its relative simplicity, the method used in BEWARE-2 to probabilistically match target to database oceanic conditions, as opposed to simple interpolation at the target condition, does provide further information on the uncertainty (confidence bands) of the runup prediction (i.e., Step 2 in Section 2.2.2 and Figure 3 in the manuscript). In this case, the simple approach therefore seems sufficient. We have included reference to this in Section 2.2.2 (lines 226–229) of the updated manuscript (see also Figure 6 at the end of this document).

The referee also makes an interesting suggestion to develop a new ML model based on the entirety of the training data (i.e., combined variation of profiles and oceanic forcing conditions), for instance through application of Radial Basis Functions (RBFs). There has been some very interesting progress made in this field in recent years. For instance, Ricondo et al. (2024) applied RBF to develop a meta-model of surf-zone hydrodynamics on reefs. In line with other existing parametric and meta-models, however, this ML model was developed for idealized reef profiles with a limited set of geometric parameters. Application of an RBF-type approach with morphologically diverse, real-world, reef profiles is still far from mathematically trivial, as the problem can be extremely ill-conditioned.

Although we do not currently consider the development of a new Neural Network (NN) meta-model to be necessary to simulate wave runup, or easily achievable for morphologically diverse reef profiles, we are providing open access to the BEWARE-2 training dataset for further research. We would be happy to support others in developing more advanced meta-models, for instance that may be able to provide estimates of more hazard indicators than wave runup such as overtopping volumes, resulting topographic change, etc.

4. The validation presented is limited to comparisons between the full-fidelity (XB-NH) model and the surrogate model. While this is the norm for many hybrid modelling studies, it would be nice to see some comparisons of the surrogate model (alongside XB-NH) to real-world observation as was done in the earlier BEWARE paper. There are lots of empirical/statistical/analytic/hybrid approaches that estimate wave runup – how much better is BEWARE-2? Given the information, it is difficult to assess how much better BEWARE-2 might be compared to these other approaches.

To the best of our knowledge, validation of a metamodel against the original model it has been trained to imitate is the norm, as also stated by the referee. The recommendation of the referee to include real-world observations (also echoed by Referee #2) is one that we fully agree with (see also Section 4.3 of the manuscript), but also one that is currently very difficult to fulfill: field

observations of wave runup on coral reef-lined beaches, particularly during energetic forcing conditions, are practically non-existent (e.g., Winter et al. 2020).

To our knowledge, the only published observations of wave runup on coral reef-lined coasts with concurrent boundary forcing conditions are presented in Quataert et al. (2020). These have a vertical resolution of approximately 1 m (limited by the individual features identified in the images), and represent an approximation of the maximum wave runup over a half hour period. As these data are necessarily quite coarse and have additionally previously been used to verify the XBeach model, we do not think it is appropriate to use these data to "validate" the BEWARE-2 metamodel.

The lack of observational data is further reflected in the validation sections of earlier metamodel studies. For instance, the original BEWARE model (Pearson et al. 2019) made use of three numerical model predictions of wave runup at Funafuti (Basilisk GN model; Beetham et al. 2015) and one empirical model estimate of wave runup at Roi-Namur (Hunt runup formulation; Cheriton et al. 2016). The HyCReWW model (Rueda et al. 2019) used the same observations as used by Pearson et al. (2019), alongside laboratory scale observations (which have in the past also been used to validate XBeach; Lashley et al., 2018), and one observation of wave runup at Lahaina (source not provided). Liu et al. (2023) similarly used numerical model simulations of wave runup at La Saline (XBeach; Bruch et al. 2020) and Roi-Namur (XBeach; Quataert et al. 2020) to validate their wave runup metamodel. To a great extent therefore, the available data of wave runup used in earlier studies are in fact laboratory scale observations that have been used to validate the XBeach model, or numerical model results of, primarily, the XBeach model.

The question whether BEWARE-2 is a better predictor of wave runup than other models is complicated by the fact that application of other metamodels to complex reef profiles is rather dependent on the subjective assessment of key reef geometry parameters, such as the reef platform width and depth, and the fore reef slope. For instance, given the observed reef profile presented in Figure 1 of this document, users of existing metamodels are required to decide what the characteristic reef platform width and depth is, which subjectively could include, or not, the reef profile from 80–180 m cross-shore position, thereby substantially affecting the prediction of wave runup. It is therefore quite tricky to objectively assess the improvement of BEWARE-2 over earlier metamodels on such complex profiles without introducing (unconscious) bias. We therefore deliberately chose to steer away from this topic in the manuscript, and instead allow for fully independent comparison of the advantages and disadvantages of the various metamodels in practical situations by the wider coastal science and engineering community.

On simpler, "idealized", reef profiles the difference in accuracy of the runup prediction of BEWARE-2 compared to other metamodels trained on XBeach-generated data (e.g., Pearson et al., 2019; Rueda et al., 2019; Liu et al., 2023) is expected to be negligible, as for these cases the metamodels have been shown to accurately mimic the results of the XBeach model.

We would finally like to state that several coauthors on this manuscript are currently involved in research projects aimed at meeting the need for field observations and full-scale laboratory measurements of wave runup on coral reef-lined coasts. We have every intention of using these field observations to scrutinize the accuracy of BEWARE-2 once the data become available and to share these findings with the coastal science and engineering community in a following manuscript once those data have been collected and analyzed.

[Figure]

*Figure 1: Example of a complex coral reef profile that is not easily described by reef platform width and depth geometric parameters.*

Abstract:

Are the unit details on verification necessary? The upper limit runup of range (20.9 m) is non-intuitive until the semi-infinite beach slope is defined in the methods section. In my view it would be better to normalise RMSE and bias and perhaps represent them as percentages for the abstract so this stated range is not needed.

Thank you for this suggestion, we have included the normalized RMSE (SI) and normalized bias to the abstract alongside the RMSE and bias (lines 14–16; see also Figure 2 at the end of this document).

A little difficult to follow… also, what is the difference with this paper and the earlier BEWARE paper (https://doi.org/10.1002/2017JC013204)?  That is front of mind to readers such as myself, who are aware of the earlier work.

We were not entirely sure what section the referee is referring to as difficult to follow. We have added explicitly the objective to provide wave runup information on morphologically diverse reef profiles in the abstract (lines 12–13; see also Figure 2 at the end of this document) and that this differs from earlier metamodels in general. We do not believe it necessary to highlight differences with specific models (i.e., Pearson et al., 2017) in the abstract.

Introduction

Ln 30 - : since publication of Hoeke et al 2013, the number of case studies attributing remotely generated swell as the primary proximal factor in island flooding events has expanded  – I

recommend adding a few more recent examples (e.g. Wadey, et al 2017, Ford et al 2018, Wandres, et al 2020, Hoeke, et al 2021) to highlight its pervasiveness among oceanic islands.

Thank you, we have included these references in lines 35–36.

Ln 64: (Pearson et al., 2017; Rueda et al., 2019; Liu et al., 2023), consider adding Beetham and Kench, 2018 to this list?

The RIOT model of Beetham and Kench (2018) is slightly different to the others originally listed here, both in model type and output information, but is certainly worth including in the overview. We have included as a "numerical model informed empirical relation" (line 67; see also Figure 3 at the end of this document).

Also, while all of these meta-modelling approaches may suffer "limited number of schematic coral reef bathymetries" how do their approaches compare to BEWARE-2? Is BEWARE-2 only better because more training data has been introduced or are there other improvements/considerations in the overall approach?

Here we refer to our response to Key Points 2 and 4 of this referee: the main objective of BEWARE-2 is to incorporate morphological diversity of reef profiles (Key Point 2) and that objective quantification of the improvement in wave runup prediction is difficult, particularly in the absence of real-world observations (Key Point 4).

Methods

Ln 94-115: I found this section circuitous and hard to follow, with poor economy of words. At the very least end Ln 98 with "… using morphological clustering technique, as summarised in the following paragraph."

We edited this section in the attempt to increase legibility. We thank the referee for his suggestion, which we have incorporated in the manuscript (see also Figure 4 at the end of this document).

Figure 2: This just looks like random coloured spaghetti – maybe sorting by mean profile steepness or runup would make this more sensible? Also, runup based on what boundary conditions? Is this normalised somehow?

The ordering of the profiles was not clear in the caption in this version of the manuscript. We have included in the caption that the profiles are ordered by mean profile steepness (profile above MSL – 15 m). We have also added that wave runup was calculated for identical wave conditions on all profiles (not normalized). In line with the suggestion by Referee #2, we have adjusted the color scheme of the figure (see also Figure 5 at the end of this document).

Ln 299 "… 5, 25, 50, 75, and 95% depth exceedance values, i.e., the depth exceeded by a given percentage of the observed profiles at each cross-shore location" not sure I understand this …

We have reworded this section to clarify (lines 308–313; see also Figure 7 at the end of this document).

Benefits and limitations and/or Conclusion sections:

I think it would be worthwhile to point out that the reef-lined coasts of many nations do not have the high resolution bathytopo information (e.g. based on LIDAR surveys) needed to make use of tools like BEWARE-2 – this paper is opportunity to point out the extremely high value of such underpinning data.

We have included this point at the end of Section 4.1 (lines 469–471; see also Figure 8 at the end of this document).

**Screenshots from updated manuscript (track-changes):**

[revised manuscript text omitted]

35 m

MSL

3500 m

**Runup rank**

lowest R$_{2\%}$        highest R$_{2\%}$

**Figure 2.** Overview of the morphology of the 195 representative reef profiles (RRPs), ordered from top left to bottom right by cross-shore distance to the 15 m depth contour ($W_{\text{reef}}$, see Section 3.2). The RRPs are color-coded according to  the relative ranking of runup simulated by XBNH for a single representative wave condition ($H_{s,0} = 5$ m, $T_p = 12$ s), with  blue indicating profiles with  relatively lower wave runup and  red those with  relatively higher wave runup.

*Figure 5: Screenshot of updated Figure 2.*

$$p_{p_{C_{DB}|C_T}}(n) = \frac{\text{GMIND}(n)}{\sum_{m=1}^{8} \text{GMIND}(m)} \tag{3}$$

The weighted nearest neighbour-type approach described above to assign probability weights to database conditions is different from the Bayesian-based interpolation applied in the BEWARE-1 meta-model. However, in Section 2.4 we will show that the relatively simple and explainable machine-learning approach applied in BEWARE-2 is easily sufficiently accurate for practical application.

*Figure 6: Screenshot of updated Methodology section with reference to difference in Bayesian approach relative to BEWARE-1.*

**2.4.1 Validation dataset**

315     The ability of the BEWARE-2 meta-process model to predict $R_{2\%}$ on morphologically diverse reefs and under varying hydro-dynamic forcing conditions is quantified using a validation dataset of 24,000 process-based, XBNH model simulations that are separate from the dataset of simulations used to train the meta-process model. To develop the validation dataset, five normal, and one to two wide, real cross-shore profiles were selected from each of the seven geographic regions (Guam, Saipan-Tinian, American Samoa, Hawai'i, Florida, Puerto Rico, and the US Virgin Islands) included in the dataset of Storlazzi et al. (2019),

320     for a total of 35 normal and 13 wide reef profiles.  Normal reef profiles representative of the  morphological diversity at every geographic region in the dataset of Storlazzi et al. (2019, i.e., 20,454 profiles in total, see Section 2.1.1) were selected  by first statistically determining  the  5, 25, 50, 75, and 95% depth exceedance values  at every cross-shore position (i.e., the depth exceeded by a given percentage of the observed profiles at  every cross-shore position; Figure 4,

325     dashed lines). Subsequently, the  observed, real-world profiles most similar to the cross-shore varying 5, 25, 50, 75, and 95% depth exceedance  values (Figure 4, solid lines) were selected for the validation dataset. Wide reef profiles were similarly selected for each geographic region from the dataset of wide coral reef profiles of Storlazzi et al. (2019, i.e., 9,712 profiles in total, see Section 2.1.1). For all geographic regions except American Samoa, the nearest observed profiles to the 25% and 75% depth exceedance of wide profiles were selected for the validation dataset. In American Samoa,

330     the only wide profile included in the database of Storlazzi et al. (2019) was selected. None of the 48 (35 normal and 13 wide reef) validation profiles were identical to the RRPs included in the training dataset.

*Figure 7: Screenshot of updated Methodology section with reference to statistical exceedance depth profiles.*

470     The XBNH model is reasonably well validated, as described in the Introduction and Methods sections, but is not perfect. A key limitation of BEWARE-2 is the one-dimensional nature of the underlying XBNH model simulations and thus the meta-process model dataset, which is unlikely to be very accurate in highly two-dimensional situations, such as in the presence of reef channels (e.g., Storlazzi et al., 2022). In addition, BEWARE-2 shows notably lower skill in reproducing XBNH runup values for very wide reef profiles (characteristic of barrier and extremely wide fringing reefs), likely due to the relatively low

475     profile coverage (only 20 RRPs) and large number of possible permutations. Although this may be improved by including a greater number of wide RRPs in the dataset, the real-world accuracy of the underlying XBNH model may likely be lower for very wide reef profiles than other reefs for which it has been validated, as one-dimensional XBNH models are unable to account for wind growth of waves or large-scale topographic refraction (Scott et al., 2020). Further extension of BEWARE-2 for these reef types is therefore currently constrained by the inherent uncertainty in applying XBNH to very wide reef profiles.

480     A final key limitation of BEWARE-2 is the underlying reef profile dataset, which is heavily biased toward data from U.S. (fringing) coral reef-lined coasts (due to consistent LiDAR data availability at the time of BEWARE-2 development) and may lack important other characteristic reef profiles such as barrier and platform reefs. Collection and dissemination of accurate, high resolution bathymetric data at other coral reef-lined coasts around the world would greatly aid application of BEWARE-2, as well as other process-based and meta-models, in global efforts to reduce the impacts of coastal flooding.

*Figure 8: Screenshot of updated Discussion section.*

**References in this document:**

Beetham, E. P., Kench, P. S., O'Callaghan, J., & Popinet, S. (2015). Wave transformation and shoreline water level on Funafuti Atoll, Tuvalu. Journal of Geophysical Research: Ocean, 120, 1–16. https://doi.org/10.1002/2014JC010472

Bruch, W., Cordier, E., Floc'h, F., & Pearson, S. G. (2022). Water level modulation of wave transformation, setup and runup over La Saline fringing reef. Journal of Geophysical Research: Oceans, 127, e2022JC018570. https://doi.org/10.1029/2022JC018570

Cheriton, O. M., C. D. Storlazzi, and K. J. Rosenberger (2016), Observations of wave transformation over a fringing coral reef and the importance of low-frequency waves and offshore water levels

to runup, overwash, and coastal flooding, *J. Geophys. Res. Oceans*, 121, 3121–3140, doi:10.1002/2015JC011231.

Lashley, C.H., Roelvink, D., van Dongeren, A., Buckley, M.L. and Lowe, R.J., 2018. Nonhydrostatic and surfbeat model predictions of extreme wave run-up in fringing reef environments. *Coastal Engineering*, 137, pp.11-27.

Pearson, S. G., Storlazzi, C. D., van Dongeren, A. R., Tissier, M. F. S., and Reniers, A. J. H. M, 2019. A Bayesian-Based System to Assess Wave-Driven Flooding Hazards on Coral Reef-Lined Coasts, *Journal of Geophysical Research: Oceans*, 122, 10 099–10 117, https://doi.org/10.1002/2017JC013204

Quataert, E., Storlazzi, C., van Dongeren, A., McCall, R., 2020. The importance of explicitly modelling sea-swell waves for runup on reef-lined coasts. *Coast Eng.* 160, 103704 https://doi.org/10.1016/j.coastaleng.2020.103704.

Ricondo, A., Cagigal, L., Pérez-Díaz, B. and Méndez, F.J., 2024. HySwash: A hybrid model for nearshore wave processes. *Ocean Engineering*, 291, p.116419.

Rueda, A., Cagigal, L., Pearson, S., Antolínez, J. A. A., Storlazzi, C., van Dongeren, A., Camus, P., & Mendez, F. J. (2019). HyCReWW: A Hybrid Coral Reef Wave and Water level metamodel. *Computersand Geosciences*, 127, 85-90. https://doi.org/10.1016/j.cageo.2019.03.004

Scott, F., Antolinez, J. A. A., McCall, R., Storlazzi, C., Reniers, A., and Pearson, S., 2020. Hydro-Morphological Characterization of Coral Reefs for Wave Runup Prediction, *Frontiers in Marine Science*, 7, https://www.frontiersin.org/articles/10.3389/fmars.2020.00361.

Winter, G., Storlazzi, C., Vitousek, S., van Dongeren, A., McCall, R., Hoeke, R., et al., 2020. Steps to develop early warning systems and future scenarios of storm wavedriven flooding along coral reef-lined coasts. *Front. Mar. Sci.* 7, 00199. https://doi. org/10.3389/fmars.2020.00199.

---

## Author Comment (AC2)

**Response to Referee #2 (Anonymous) comments on**

*Rapid simulation of wave runup on morphologically diverse, reef-lined coasts with the BEWARE-2 meta-process model*

*Robert McCall, Curt Storlazzi, Floortje Roelvink, Stuart Pearson, Roel de Goede, and José Antolínez*

We would like to thank the anonymous referee for their suggestions and constructive comments on this manuscript. We have attempted to address the points made by the referee in the updated manuscript and/or have provided our rebuttal below. In the following, the referee's comments are given in **black** font and our response in **blue** font.

We have been advised that we are not able to upload the updated manuscript to the discussion portal at this time. In lieu of providing the referee with an updated copy of the manuscript, we have therefore included screenshots of changes to the manuscript, where appropriate, at the end of this document.

General comments:

This paper is concerned with the development and application of a meta-process modelling system to address the need for a fast, robust prediction of runup on reef-lined coasts. The scientific significance of the paper is substantial given that it addresses a very real problem associated with the need to better predict coastal flooding along reef-lined coasts. The use of large data sets which are validated against the results of a numerical model (XB-NH), and the incorporation of roughness variations make this an important contribution. The paper is generally well written and well presented. Progress towards an early warning system for such vulnerable areas would be highly beneficial.

We thank the referee for their support of the topic addressed in this manuscript.

Specific comments:

Though the paper is very thorough and uses large sets of data, as the authors suggest, there is a skewed focus on U.S. data, and testing the model with examples from other locations would be interesting to see.

We fully agree with the referee, as previously noted in our Section 4.1 (Next steps). Stating that, it is the range of morphologies of the reefs (fringing or atoll reefs versus barrier reefs, each of which have thousands of samples and extend over scales of meters to 10s of kilometers in the cross-shore) spread across two oceans that matter more than the nationality of those reefs, and we feel we have a good first pass at them. But as also noted previously, more reef morphologies can always be added to the database and new RRPs developed to expand the metaprocess model's database.

The validation against the XB-NH runup values is understandable given the complexity involved with obtaining field measurements, however it would be interesting to see a comparison with field data, even if only for a very limited number of scenarios. The use of a 1D model is certainly far more practical, however would validation against a small set of scenarios with field or physical model data help to reduce the uncertainty as to the extent of these effects on the runup values?

This comment is broadly in line with a comment made by Referee #1, and we partly refer back to our response to that comment regarding the availability of real-world observations of wave runup on coral reef-lined coasts with which to assess the skill of BEWARE-2. Our conclusion in our response to Referee #1 is that these data are simply not available; previous metamodel studies have instead mainly used the wave runup results of numerical models (primarily

XBeach) that have been validated for wave transformation as a proxy for true observations. In effect we are doing the same in this manuscript, as we start off by training the metamodel with the model validated for wave transformation in field conditions (XBeach-NH+), and subsequently compare BEWARE-2 wave runup predictions to runup predictions of the validated model (XBeach-NH+).

Specifically regarding the use of laboratory data: we know from previous research (listed in Section 2.1.2 of the manuscript) that XBeach-NH+ is well able to simulate laboratory scale wave transformation and even wave runup (Lashley et al. 2018). However, translation of these results to field cases is here also limited by the 1D (2DV) assumption of the wave flume: alongshore gradient effects, including large-scale reef platform circulation and its effect on setup, are unfortunately not captured in these physical model experiments. To assess the accuracy of the BEWARE-2 metamodel for these cases, we will still require field observations.

We would like to reiterate from our response to Referee #1 that we very much intend to assess the skill of BEWARE-2 once new observational data become available.

I agree with the comments made by RC1, that a slightly more detailed/clearer explanation of the differences in the methods used in BEWARE2 compared to BEWARE would be beneficial to the paper.

This suggestion was indeed also made by Referee #1 and we refer to our response to our response to Referee #1 on this point (Key Point 2).

Figure 2. I think this figure could be improved. I understand the use of the 195 RRPs but is there a deliberate order to the way they are presented? Could this be improved? Is yellow the best choice of colour for the low runup?

This suggestion was also given by Referee #1 and we have followed the suggestion of Referee #2 to change the color scheme of this figure.

Could the font size be increased in Figure 3 and 5?

Where figure spacing allows, we have increased font size of these figures. We would like to note that for online readers of the manuscript, all figures are in vector format to support zooming in to any section of the figure. We have included a screenshot of the updated Figure 3 and 5 at the end of this document. Note that these screenshots are not in vector format, those in the manuscript are in vector format.

Figure 4: Is it necessary to include all profiles? The grey can barely be seen when printed.

We have included the grey lines to indicate the entire spread of observed profiles, but not with the intention for readers to study the individual profiles themselves. We have included a message to this effect in the figure caption.

I may have missed this, but what were the computer specifications used to run the XB-NH simulations?

For the purpose of the comparison of computation times between XBNH and BEWARE-2 (i.e., lines 448–449), is based on simulations on a 12th Gen Intel Core i5 laptop (referred to as a "standard desktop computer" in the manuscript) but should be fairly consistent for any "standard" computer. In the manuscript we do not specify the hardware used to develop the XBNH training dataset. Due to the 1D nature of the XBNH simulations, generating the training dataset is possible using separate or clusters of desktop PCs.

Technical corrections:

Ln 191 "…converted on…" should this be "…converted into…" ?

Thank you, we have corrected this in the manuscript.

Ln 472 "…"the influence reef health…" should this be "…the influence of reef health…"?

Thank you, we have corrected this in the manuscript.

Ln 474 "100s years…" should this be "…100s of years…"?

Thank you, we have corrected this in the manuscript.

**Screenshots from updated manuscript (track-changes):**

[Figure]

*Figure 1: Screenshot of updated Figure 3.*

[Figure]

**Figure 4.** Observed, real-world, normal (gray) and wide (dark blue) cross-shore reef profiles for each of the seven geographic regions included in the dataset of Storlazzi et al. (2019), indicating the natural variability in reef morphologies. Colored dashed lines indicate the cross-shore varying, statistical 5, 25, 50, 75, and 95% depth exceedance  values. Colored solid lines indicate  the real-world profile  most similar to each depth exceedence, which were used for model validation. For the wide reefs, only the 25 and 75% depth exceedance profiles were extracted for model validation.

*Figure 2: Screenshot of updated caption for Figure 4.*

[Figure]

*Figure 3: Screenshot of updated Figure 5.*